# Double-strand breaks induce inverted duplication chromosome rearrangements by a DNA polymerase δ-dependent mechanism

Amr M. Al-Zain [1,2], Mattie R. Nester[2], Iffat Ahmed[2] & Lorraine S. Symington [2,3] ✉

Inverted duplications, also known as foldback inversions, are commonly observed in cancers and are the major class of chromosome rearrangement recovered from yeast cells lacking Mre11 nuclease activity. Foldback priming at DNA double-strand breaks (DSBs) is one mechanism proposed for the generation of inverted duplications. However, the other pathway steps have not been fully elucidated. Here, we show that a DSB induced near natural inverted repeats drives high frequency inverted duplication in Sae2 and Mre11-deficient cells. We find that DNA polymerase δ proof-reading activity, but not Rad1 nuclease, trims the heterologous flaps formed after foldback annealing. Additionally, Pol32 is required for the generation of inverted duplications, suggesting that Pol δ catalyzes fill-in synthesis primed from the foldback to create a hairpin-capped chromosome that is subsequently replicated to form a dicentric inversion chromosome. Finally, we show that stabilization of the dicentric chromosome after breakage involves telomere capture by non-reciprocal translocation mediated by repeat sequences or by deletion of one centromere.

Most human cancer cells exhibit genomic instability, ranging from elevated mutation rates to gross chromosome rearrangements and aneuploidy. Gross chromosome rearrangements (herein called GCRs) include translocations, deletions, inversions and amplifications. The prevailing view is that GCRs are generated through error-prone processing of damaged chromosomes[1]. Although the nature of the initiating lesions for GCRs is unknown, much of the genetic evidence from yeast and human cells implicates DNA double-strand breaks (DSBs) formed directly or indirectly by problems during DNA replication[2].

GCRs have been studied extensively in yeast using genetic assays developed by the Kolodner group[3,4]. The classical GCR assay detects the spontaneous loss of two counter-selectable genes, *CAN1* and *URA3*, that are present on the left arm of chromosome V (Chr V-L), telomeric to the last essential gene. Using this assay, the rates of GCRs in various genetic backgrounds as well as the types of rearrangements have been

well characterized. In wild-type (WT) cells, the most common type of rearrangement is telomere addition, in which terminal Chr V-L loss is accompanied by de novo addition of sequences telomeric to the break site[2,3]. Other rearrangements include interstitial deletions, non-reciprocal translocations and inverted duplications[1].

Inverted duplications, also called foldback inversions or palindromic duplications, are thought to arise from a DSB intermediate[5–8]. There are several proposed models for the formation of inverted duplications initiated by DSBs and these can be broadly categorized as replicative or non-replicative. Non-replicative inverted duplications arise through fusion of sister chromatids after replication of a chromosome that has sustained a DSB or telomere attrition during G1 phase[9–11]. Sister-chromatid fusions occur mainly by microhomology-mediated end joining[12–14], but have also been reported to occur by single-strand annealing (SSA) involving long inverted repeats[15].

[1]Program in Biological Sciences, Columbia University, New York, NY 10027, USA. [2]Department of Microbiology & Immunology, Columbia University Irving Medical Center, New York, NY 10032, USA. [3]Department of Genetics & Development, Columbia University Irving Medical Center, New York, NY 10032, USA. ✉ e-mail: lss5@cumc.columbia.edu

Telomere-telomere fusions, in contrast, are largely dependent on non-homologous end joining (NHEJ)[14,16–18]. Replicative inverted duplications are thought to occur via intra-strand annealing between inverted repeats (IRs) exposed by end resection[19–23]. Subsequent fill-in synthesis and ligation create a hairpin-capped chromosome. It has been suggested that during the next cell cycle, the replisome loops at the hairpin and replicates back to the end of the chromosome. Both sister-chromatid fusions and replication through hairpin-capped chromosomes create dicentric inversion chromosomes.

Dicentric chromosomes are unstable as they can be broken during cytokinesis when they are pulled apart by each daughter cell and form a bridge. Asymmetric breakage results in an inverted duplication, the degree of which depends on the location of the break. In budding yeast, breakage occurs either at the center of telomere-telomere fusions or within a 25–30 kb region near the centromere of dicentrics without telomere fusions[24,25]. Broken dicentrics can then undergo sister-chromatid fusion, initiating a breakage-fusion-bridge cycle, or be stabilized by acquisition of a telomere either by de novo telomere addition or by break-induced replication (BIR) through a repeat sequence[26,27]. Alternatively, a dicentric can be stabilized by loss of one of its centromeres[26,28].

The frequency of inverted duplications detected using the spontaneous GCR assay is elevated in *sae2Δ* mutants as well as cells defective for Mre11 nuclease activity[5–7]. At the center of those inverted duplications are naturally occurring 3–15 bp long IRs[5–7], suggesting that they form via intra-strand foldback annealing between the IRs. Mre11 (as part of the Mre11-Rad50-Xrs2 complex) has endonuclease activity that is stimulated by Sae2 and is proposed to cleave hairpin-capped ends[29–33]. Mre11 and Sae2 are thus thought to prevent inverted duplications by resolving the hairpin-capped chromosome intermediate[5–8,30,34,35].

Studies have shown that a DSB near long artificially integrated palindromes or quasi palindromes (40 bp or longer) can lead to inverted duplications through a hairpin-capped intermediate[15,23,36,37]. DSBs near short IRs have also been shown to simulate the formation of inverted duplications[7,8]. However, the specific effect of the DSB on the frequency and the mechanism of GCRs has not been systematically studied. In this study, we monitored the repair outcome of a CRISPR/Cas9-induced DSB near naturally occurring IRs. Our data show that inverted duplications occur at a surprisingly high frequency in cells deficient for Mre11 nuclease activity. Similar to previously proposed models, the inverted duplications occur through intra-strand foldback annealing at resected IRs to form a hairpin-capped chromosome that is a precursor to dicentric inversion chromosomes. We identify two roles for DNA polymerase δ in the generation of foldback inversions. First, the proofreading activity is required to remove heterologous flaps formed during foldback annealing, and second, the Pol δ processivity subunit, Pol32, is important for fill-in synthesis to generate a hairpin-capped chromosome. In addition, we find that stabilization of dicentric chromosomes after breakage occurs by Rad51-dependent recombination between repeat sequences or by deletion of one centromere.

## Results

### Cells lacking Mre11 endonuclease activity exhibit increased survival to a DSB induced near natural inverted repeats

To determine whether a targeted DSB induces inverted duplications, particularly in the context of Mre11 nuclease deficiency, we used CRISPR/Cas9[38] to create a DSB near a naturally occurring IR on Chr V-L. The sequence chosen was observed to be at the center of four spontaneous inverted duplications previously recovered from *sae2Δ* cells[6] (Fig. 1a). It is an imperfect repeat of 11 bp with two mismatches, separated by 6 bp, within the *CAN1* gene. A gRNA (hereafter referred to as gRNA-17) was designed to target a sequence 17-bp telomeric to the IR, thus creating a DSB 20 bp from the IR as Cas9 cuts 3 bp from the target sequence 3′ end. Sequences telomeric to the DSB are non-essential for

cell viability, permitting recovery of a variety of GCRs. The gRNA-17 expression cassette was stably integrated into the genome.

We initially used the galactose-inducible *GAL1* promoter to drive expression of Cas9 but observed a high level of leaky expression when cells were grown under non-inducing conditions, as measured by inactivation of *CAN1* in the presence of gRNA-17 (Supplementary Fig. 1a). To reduce background cleavage by Cas9, we switched to an estrogen-inducible LexA transcription factor fusion (LexA-ER-AD) to drive expression of Cas9 under the control of the *lexO* operator and a minimal $P_{CYC1}$ promoter[39] (Supplementary Fig. 1b). However, we still observed a high *CAN1* mutation rate under non-inducing conditions (Supplementary Fig. 1a). To further reduce background cleavage by Cas9, we fused an ER domain to Cas9 (*lexO-Cas9-ER*). Using this system, Cas9 expression as well as nuclear localization is dependent on the addition of β-estradiol to the medium. A single cassette containing the transcription factor and *lexO-Cas9-ER* was stably integrated into the genome. Due to the presence of two inverted ER domains in the construct, there is a possibility for spontaneous recombination between them, which would disrupt the LexA-ER-AD transcription factor and eliminate Cas9 expression. Although these events were expected to be rare, they represent about 20% of the total number of colonies that grow on inducing medium in WT cells and were filtered out from subsequent analysis.

Logarithmically growing haploid cells were plated on media ± β-estradiol and cell survival was determined by the ratio of colonies that grew on β-estradiol-containing medium versus medium lacking β-estradiol. Since Cas9 is constitutively expressed when cells are plated on medium containing β-estradiol, and there is no repair template for homologous recombination, cells can only grow if they lose the gRNA target sequence. Therefore, repair in surviving cells is likely to be inherently mutagenic. The gRNA target sequence can be lost by indels via NHEJ or larger scale sequence loss (Fig. 1a). The survival frequency of WT cells was 0.04% ± 0.07 (Fig. 1b), an order of magnitude lower than the survival of cells to DSBs generated by HO or I-SceI endonucleases[40,41], suggesting that NHEJ is ineffective in repair of the Cas9-induced DSB. Remarkably, survival of *sae2Δ* cells (8.1% ± 10.7) was ~200-fold higher than that of WT cells (Fig. 1b, Supplementary Table 1). The survival frequencies of *sae2Δ*, *mre11Δ* and *mre11-H125N* (deficient for the nuclease activity of Mre11) mutants were similar, suggesting more efficient stabilization of the broken chromosome than in WT cells.

The different potential repair outcomes are mutagenic NHEJ or chromosome rearrangements. To distinguish between these possibilities, we used PCR to detect the presence of 250 bp on either side of the break (Fig. 1a, primer pair P3/P4). A PCR product indicates repair of the break without extensive loss of sequences, likely due to inaccurate NHEJ. PCR analysis of several individual clones derived from WT cells revealed that about 16% of survivors retained sequences surrounding the cut site (Fig. 1c). Sequencing of P3/P4 PCR products revealed the presence of indels or base substitutions at the gRNA target site clustered in the first two nucleotides, or short deletions, indicative of NHEJ (Supplementary Fig. 2a). One possible explanation for the inefficiency of NHEJ is that Cas9-generated DSBs have mostly blunt ends[38,42], which are poorly ligated by the NHEJ machinery in budding yeast[43,44]. In vivo studies imply that the ends generated by Cas9 can have 1–2 nt overhangs since the repair products often exhibit templated insertions, but this is influenced by the sequence context of the DSB[45–47]. Although *SAE2* deletion has been shown to increase the frequency of NHEJ[48,49], very few of the *sae2Δ* survivors had a P3/P4 PCR product (Fig. 1c). Therefore, the significant increase in the survival frequency of those cells reflects increased channeling to a repair pathway that leads to sequence loss.

To further characterize the mutagenic repair outcomes, we screened survivors by PCR using sub-telomeric primers (Fig. 1a, P1/P2) to determine whether the terminal sequences of Chr V-L were retained.

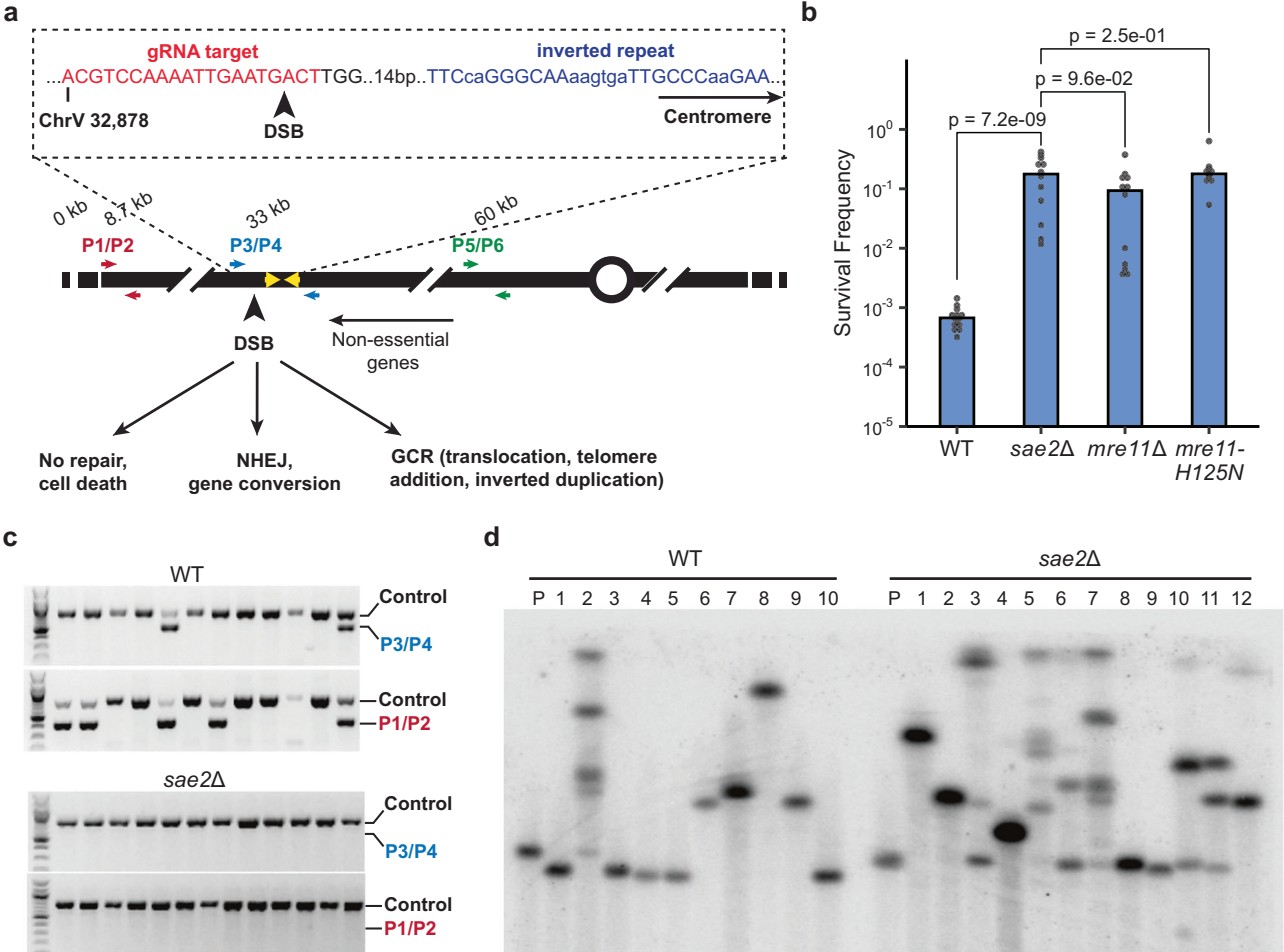

**Fig. 1 | Cells lacking Mre11 endonuclease activity exhibit increased survival to a DSB induced near a natural inverted repeat. a** Top: Sequence of the gRNA target (red font) located telomere proximal to the targeted IR (blue font). The mismatches within the 11 bp imperfect repeats and the 6 bp spacer are shown in lower case. Bottom: Schematic of the left arm of Chr V showing the location of the DSB and inverted repeats (yellow arrows). Primer pairs to detect retention of terminal sequence, NHEJ events, and copy number increase are shown in red, blue and green, respectively. **b** Survival frequencies of the indicated strains in response to expression of Cas9 and gRNA-17 (see the "Methods" section). *P* values were determined using a two-tailed *t* test. WT, *n* = 14; *mre11*Δ, *n* = 12; *mre11-H125N*, *n* = 12; *sae2*Δ, *n* = 12; 3–5 biological replicas. **c** Twelve independent clones from WT and *sae2*Δ cells were analyzed using P3/4 and P1/2 primers to detect NHEJ events or loss of Chr V terminal sequence. **d** PFGE of 10 WT clones that were pre-screened to eliminate NHEJ and homeologous gene conversion events, and 12 *sae2*Δ clones analyzed directly from β-estradiol plates. Shown is a Southern blot with a probe that hybridizes to *PCM1* on Chr V. Source data are provided as Source Data Fig. 1.

Clones in which the terminal sequences are retained, but not the cut site sequences (three shown in Fig. 1c) may be indicative of interstitial deletions. Alternatively, sequences that anneal to the cut site primer could be altered without otherwise significant sequence loss. To distinguish between the two outcomes, we used primers 200 bp further upstream and downstream of the cut site primers to screen survivors that only yielded the terminal fragment PCR product (Supplementary Fig. 2b, P7/P8). P7/P8 gave a product in all eleven tested WT survivors that retained Chr V-L terminal PCR product but not the P3/P4 PCR product (Supplementary Fig. 2c). Sequencing of those PCR products indicates that repair occurred by gene conversion using the *LYP1* gene on Chr XIV as a template (Supplementary Fig. 2d). *LYP1* encodes a lysine permease that has only 61.6% DNA sequence homology to *CAN1* and was not expected to template homology-directed repair at a detectable frequency[50–52]. In a previous study, rare rearrangements between the highly diverged *CAN1* and *LYP1* genes were detected only in the absence of Sgs1 helicase[53], whereas here they were detected in 40% of the WT clones (Figs. 1c and 2c).

To determine the size of Chr V in WT and *sae2*Δ clones, we performed pulsed-field gel electrophoresis (PFGE) of intact chromosomes and probed for Chr V (Fig. 1d). The WT clones for PFGE analysis were pre-screened by PCR using P3/4 and P7/8 primer sets to eliminate those resulting from repair by NHEJ or homeologous gene conversion, whereas the *sae2*Δ clones were not pre-screened. Compared to the parental strain, Chr V of WT survivors was of aberrant size. Half exhibited chromosome truncations (#1, 3, 4, 5 and 10 in Fig. 1d), characteristic of telomere addition or interstitial deletions. The remainder exhibited chromosome expansions, likely due to translocations or inverted duplications. One clone exhibited multiple bands (#2 in Fig. 1d), indicative of a heterogenous population of cells that have undergone different rearrangements. Most survivors from *sae2*Δ cells exhibited chromosome expansions, some of which have multiple bands that hybridize with the Chr V probe, reminiscent of *sae2*Δ inverted duplication clones recovered using the classical GCR assay[5,6,26,54].

## Sae2 suppresses DSB-induced inverted duplications

To test whether the GCRs in the *sae2*Δ clones that survived DSB formation are due to inverted duplications, genomic DNA from twelve survivors was analyzed by restriction endonuclease digestion and Southern blotting (Fig. 2a). Restriction digestion of DNA with an inverted duplication would result in bands that are twice the size of the

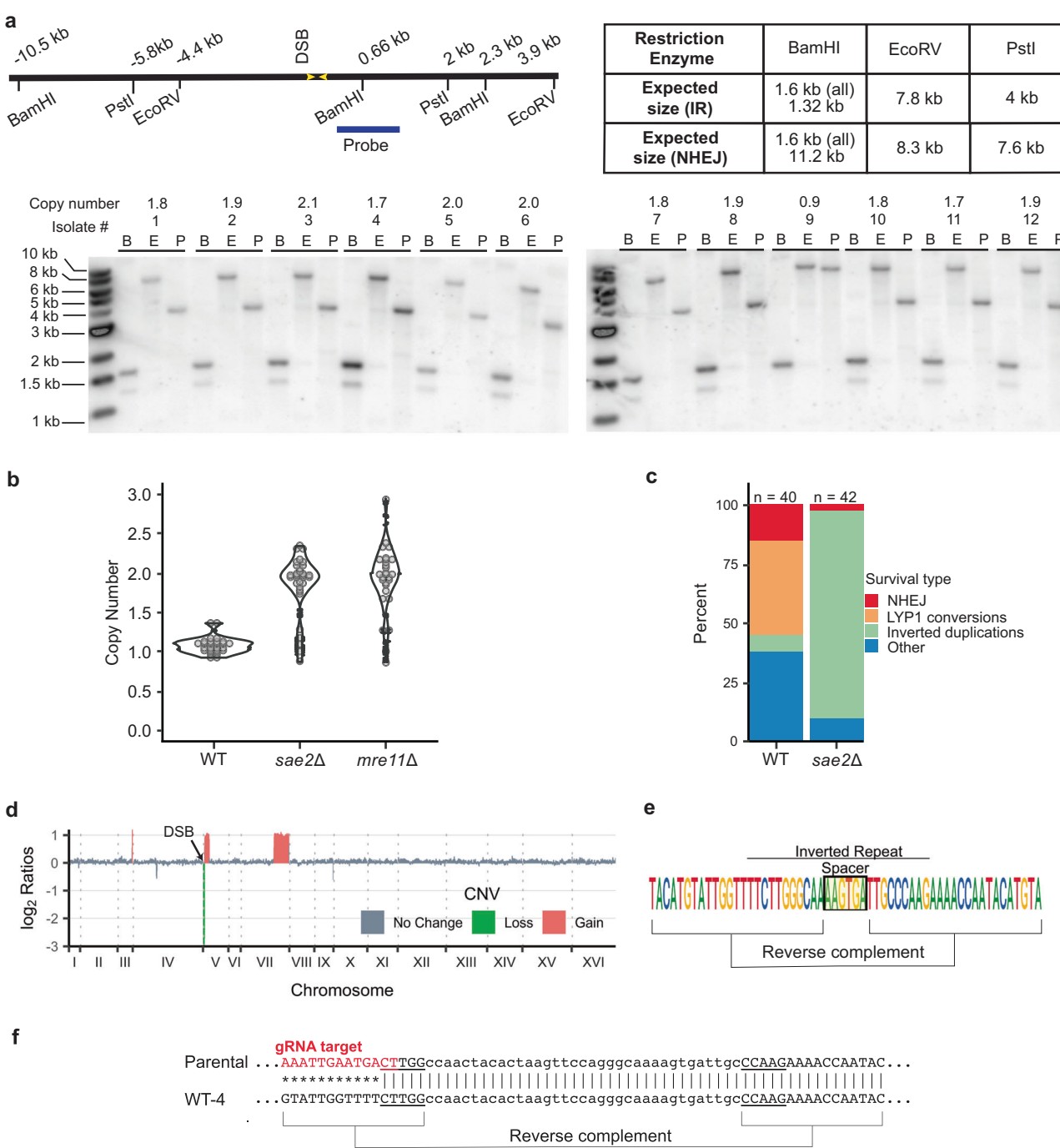

**Fig. 2 | Mre11 endonuclease activity suppresses inverted duplications. a** The schematic indicates the distance of restriction endonuclease sites from the IR and expected sizes of fragments from inverted duplication clones. Genomic blot of DNA isolated from 12 independent *sae2Δ* clones from one trial and digested with BamHI, EcoRV or PstI. The copy number of sequences detected by P5/6 primers, as determined by qPCR, are shown above each clone analyzed. **b** Independent clones derived from WT, *sae2Δ* or *mre11Δ* strains were analyzed by qPCR (P5/6 primers) to detect increased copy number of sequence between the target IR and centromere (see the "Methods" section). **c** Distribution of survivor types in WT and *sae2Δ* cells plotted as percent of total events analyzed. **d** WGS of a *sae2Δ* clone showing sequence loss centromere distal to the targeted DSB, sequence duplication centromere proximal to the DSB, and a secondary event involving partial duplication of Chr VII. **e** The *sae2Δ* clones analyzed had all corrected the mismatches present in the imperfect IR to generate a perfect palindromic duplication. **f** The center of inverted duplication detected in two wild-type clones. Top: the inverted repeat from which the inverted duplication initiated. Underline is the inverted repeat and in lower case the spacer between repeats. Bottom: the sequence at the center of the inverted repeated as detected by WGS; * indicates sequence lost centromere distal to the IR. Source data are provided as Source Data Fig. 2.

distance of the restriction site from the inverted duplication center. Eleven of twelve clones analyzed had restriction fragments consistent with the presence of an inverted duplication (all except for #9 in Fig. 2a). The remaining clone had repaired the DSB by NHEJ. In addition, we used a real time PCR (qPCR) strategy to screen a larger number of clones for copy number change in a region centromeric to the cut site (Fig. 1a, primers P5/P6) relative to a control locus in a different chromosome (*ADH1*). The majority of *sae2Δ* and *mre11Δ* survivors had an increased copy number compared with WT (Fig. 2b), indicative of duplications and consistent with the restriction digestion analysis. The

**Table 1 | Target IR and repeat elements used for secondary rearrangements of inverted duplications**

| Genotype | IR used | | | Secondary event repeat element | | | Total |
|---|---|---|---|---|---|---|---|
| | Target IR | PAM IR[a] | Other IR | *Delta* | *PAU* | Other | |
| *sae2Δ* | 11 | 0 | 0 | 7 | 3 | 1[b] | 11 |
| *sae2Δ rad1Δ* | 8 | 0 | 0 | 4 | 1 | 3[c] | 8 |
| *sae2Δ pol3-01* | 9 | 9 | 1 | 13 | 3 | 3[d] | 19 |
| *sae2Δ rad1Δ pol3-01* | 6 | 11 | 1[e] | 5 | 4 | 9[f] | 18 |
| *sae2Δ rad51Δ* | 9 | 0 | 1 | 8 | 2 | 0 | 10 |
| *sae2Δ pol32Δ* | 5 | 0 | 1 | 4 | 1 | 1[g] | 6 |
| WT | 0 | 2 | 1 | 3 | 0 | 0 | 3 |

[a]5-bp-long inverted repeat that includes the PAM site, see Fig. 2f and Supplementary Fig. 4b.
[b]Complex rearrangement.
[c]Two complex rearrangements and one telomere addition event.
[d]Two complex rearrangements and one with a mixed population.
[e]Population: mixture of target IR and PAM IR.
[f]Seven inversion chromosomes with deletion of one centromere, one telomere addition event, and one translocation involving a tRNA.
[g]Inversion chromosome with deletion of one centromere.

qPCR and Southern blot analyses revealed that most of the *sae2Δ* survivors have inverted duplications, whereas this class of GCR is rare in WT cells (Fig. 2c).

To identify the DNA sequences at the centers of inverted duplications, we prepared whole genome deep sequencing libraries from inverted duplication clones recovered from *sae2Δ* cells. Illumina paired-end sequencing revealed copy number changes consistent with an inverted duplication initiated at the Cas9-induced DSB. Sequences telomeric to the DSB were lost, whereas a duplication was detected centromeric to the DSB (Fig. 2d). Furthermore, most of the *sae2Δ* clones exhibited an additional duplication of sequences from another chromosome; these are described in detail later. We were able to obtain the sequence at the breakpoint for the inverted duplications using Comice, which is part of the Pyrus software suite[5], and by assembly of discordant read pairs of which one read maps to the vicinity of the DSB site. For all *sae2Δ* clones sequenced, the inverted duplication was centered on the natural IR targeted by gRNA-17 (hereafter referred to as the target IR), consistent with the IR driving inverted duplications (Fig. 2e, Table 1). Interestingly the two mismatches in the 11 bp imperfect inverted repeat were corrected to match the centromere proximal repeat. This finding suggests that the 3′ end to be extended after forming a foldback undergoes polymerase proofreading or mismatch repair. Surprisingly, we found that two of three inverted duplications recovered from WT cells used a different 5-bp-long IR that is located only one bp from the DSB and has a spacer of 35 bp (Fig. 2f), while the other had a complex rearrangement at its center. The 5-bp-long IR encompassing the gRNA-17 PAM site is referred to as the PAM IR (Table 1). These findings are consistent with previous studies indicating reduced Mre11 endonuclease activity towards hairpins with long (>12 nt) spacers[7,35,55].

The high survival frequency of the *sae2Δ* mutant and biased recovery of inverted duplications could potentially be a consequence of the stability of the Cas9 post-cleavage complex[56]. To address this concern, we replaced the binding site for gRNA-17 with a 36-bp-long HO cleavage site, which was designed to create a DSB 20 bp from the target IR, the same distance as generated by Cas9 (Supplementary Fig. 3a). Because the DSB produced by HO can be repaired by NHEJ[41], we obtained the expected higher frequency of cell survival in the WT strain (0.33%) (Supplementary Fig. 3b, Supplementary Table 1). Of these survivors, around 80% had repaired the DSB by NHEJ (Supplementary Fig. 3c). Remarkably, survival of *sae2Δ* cells was almost 100-fold higher, reaching 26%. Similar to our observations with Cas9, the majority of *sae2Δ* cells surviving HO induction formed small colonies indicative of inverted duplications[6,34] and qPCR analysis confirmed this prediction (Supplementary Fig. 3d). Thus, the large increase in

DSB-induced inverted duplications observed in the *sae2Δ* mutant reflects aberrant processing of a DSB and is not a consequence of Cas9 retention.

Overall, these data indicate that a DSB near an inverted repeat is sufficient to induce a high frequency of inverted duplications in *sae2Δ* and *mre11Δ* backgrounds. Furthermore, they suggest that mutagenic repair that leads to GCRs is strongly suppressed in WT cells, with the nuclease activity of Mre11 playing a major role within the context of a DSB near short, inverted repeats.

### Inverted repeats proximal to a DSB are necessary for the generation of inverted duplications

To confirm that the target IR is important for the generation of inverted duplications, we scrambled the genomic sequence corresponding to the IR by CRISPR-Cas9-mediated gene editing, leaving the sequence targeted by gRNA-17 intact (Supplementary Fig. 3e). Mutation of the IR reduced the survival of *sae2Δ* cells by about 20-fold (Fig. 3a, Supplementary Table 1). Moreover, only 16% of the survivors had inverted duplications, compared to 91% of *sae2Δ* cells with the natural inverted repeat (Fig. 3b, c). The inverted duplications in the strain with the scrambled IR initiated from other inverted repeats (Supplementary Fig. 3f), in agreement with a previous study showing use of alternate IRs when the target IR is deleted[7]. Interestingly, 34% of the survivors with the scrambled IR had P3/4 PCR products indicative of repair by NHEJ (Fig. 3c and Supplementary Fig. 3g), consistent with previous observations that *sae2Δ* increases the frequency of NHEJ[48,49]. Overall, these results suggest that an IR proximal to a DSB is required for formation of inverted duplications, likely functioning by forming a hairpin-capped chromosome via a foldback mechanism.

### The proofreading activity of DNA Pol δ contributes to heterologous flap removal

The initiating DSB is located 20 bp from the IR; therefore, formation of a hairpin-capped chromosome would require cleavage of a 20 nucleotide (nt) long heterologous flap (Fig. 4a). The Rad1–Rad10 complex has been shown to cleave 3′ heterologous flaps generated during recombination[57,58]. Heterologous flaps of 20 nt or less can also be removed by the 3′-5′ proofreading exonuclease activity of DNA polymerase δ[59]. Thus, we tested the requirement for Rad1 and DNA Pol δ in the formation of inverted duplications. Cells lacking *SAE2* and *RAD1* survive the DSB to a similar degree as *sae2Δ* cells (Fig. 4b), and most of the survivors tested exhibited duplications (Fig. 4c). This finding suggests that Rad1–Rad10 is dispensable for the formation of inverted duplications when the flap is 20 nt long, consistent with a previous report on its activity on HR substrates with flaps of a similar

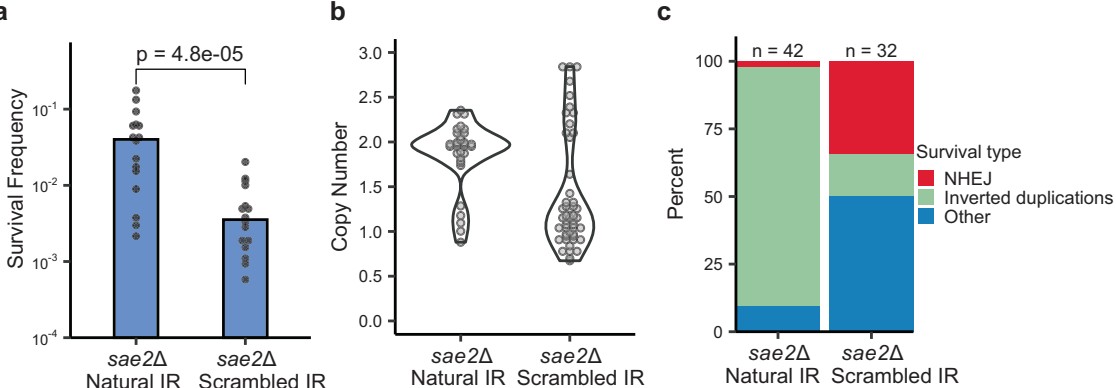

**Fig. 3 | The target IR is required for inverted duplications. a** Survival frequencies of *sae2Δ* strains with the natural or scrambled IR in response to expression of Cas9 and gRNA-17. *P* value was determined using a two-tailed *t* test. *sae2Δ* natural IR, *n* = 16; *sae2Δ* scrambled IR, *n* = 16; 5 biological replicas. **b** Independent clones of the indicated genotype were analyzed by qPCR to detect increased copy number of sequence between the target IR and centromere. **c** Distribution of survivor types in *sae2Δ* cells with the natural IR or scrambled IR plotted as percent of total events analyzed. Source data are provided as Source Data Fig. 3.

length[59]. In contrast, *sae2Δ pol3-01* cells, which have a defect in the catalytic activity of the proofreading domain of Pol δ[60], showed significantly lower survival than *sae2Δ* and *sae2Δ rad1Δ* mutants and reduced duplications in the survivors (Fig. 4b, c, Supplementary Table 1). Furthermore, *sae2Δ pol3-01 rad1Δ* cells showed similar survival to *sae2Δ pol3-01*, suggesting a non-redundant role for proofreading activity of Pol δ for inverted duplication formation. Consistent with this conclusion, 27% of clones from *sae2Δ pol3-01* survivors retained sequences surrounding the cut site, indicative of NHEJ (Supplementary Fig. 4a). By contrast, all the clones analyzed from *sae2Δ rad1Δ* cells had lost sequences surrounding the cut site. The inverted duplication defect in the *sae2Δ pol3-01* strain is surprising, since it was previously reported that Rad1–Rad10 can substitute for the proofreading activity of Pol δ during flap removal when the flaps are <20 nt long[59]. We additionally tested the effect of eliminating the Mus81-Mms4 endonuclease but did not observe a decrease in the frequency of survivors or change in the fraction of events with inverted duplications relative to the *sae2Δ* single mutant (Supplementary Table 1).

Deep sequencing of 19 inverted duplication clones from the *sae2Δ pol3-01* double mutant revealed that 9 had used the target IR and 9 used the PAM IR that would generate a heterologous flap of only 1-nt (Supplementary Fig. 4b, Table 1). The other clone used a different IR centromere proximal to the target IR. Thus, both the frequency and the spectrum of events in the *sae2Δ pol3-01* double mutant is changed relative to the *sae2Δ* single mutant. Eleven of 18 inverted duplication clones from the *sae2Δ pol3-01 rad1Δ* triple mutant used the PAM IR indicating that IR usage is not significantly altered as compared to the *sae2Δ pol3-01* double mutant (Table 1). By contrast, all eight of the *sae2Δ rad1Δ* inverted duplications sequenced were centered on the target IR. These data confirm the importance of Pol δ proofreading activity in trimming heterologous flaps.

The defect in forming inverted duplications in the *sae2Δ pol3-01* mutant could be a consequence of the mismatches present in the target IR. Following foldback annealing, there is a 2-bp mismatch 3 bp away from the end of the repeat. If heterologous flap cleavage occurs at the end of the repeat, the remaining mismatches may present an obstacle for fill-in synthesis if not corrected and could contribute to the requirement for Pol δ proofreading activity. Consistent with this idea, Pol δ proofreading activity removes mismatches close to the 5′ invading end during homology-directed repair[50,61]. The sequencing data indicate that these mismatches were corrected in all *sae2Δ* inverted duplication clones, except for two clones from the *sae2Δ pol3-01* double mutant. To determine whether the reduced frequency of inverted duplications in *sae2Δ pol3-01* cells is due to a defect in

mismatch correction rather than in heterologous flap cleavage, we modified the chromosomal sequence to perfect the target IR (Supplementary Fig. 4c). The survival frequency of *sae2Δ* cells with the perfect inverted repeat was slightly increased relative to the original inverted repeat but survival and inverted duplications were still largely dependent on Pol δ proofreading activity (Fig. 4b, c, Supplementary Table 1). Thus, the *pol3-01* defect is due to its role in heterologous flap removal and correction of mismatches within the foldback likely occurs by the canonical mismatch repair pathway with a minor contribution from Pol δ proofreading activity.

Increasing the distance between the DSB and the IR is expected to decrease the frequency of inverted duplications if removal of long, heterologous flaps is inefficient or if a long flap destabilizes base-pairing between the repeats[7]. We designed two additional gRNAs that target sequences that are 48 and 160 bp away from the inverted repeat (Supplementary Fig. 4d). As predicted, the survival frequency of *sae2Δ* cells decreased as the DSB distance from the IR was increased (Fig. 4d) and fewer inverted duplications were recovered (Fig. 4e). This finding supports the hypothesis that the inverted duplications observed initiate by foldback intra-strand annealing at the IR. Although survival of *sae2Δ rad1Δ* cells using gRNA-48 or gRNA-160 was not significantly different to the *sae2Δ* single mutant, there was a decrease in the number of inverted duplications in *sae2Δ rad1Δ* cells using gRNA-160 (*p* = 0.002) (Supplementary Table 2). To determine whether events scored as inverted duplications had initiated at a different IR to the target IR, we used a PCR strategy to detect sequences located between the binding site for gRNA-17 and the target IR (see the "Methods" section for details). The fraction of inverted duplications centered on the target IR decreased as the distance of the DSB from the target IR increased, and this effect was accentuated in the *rad1Δ* derivative using gRNA-48 (*p* = 0.013) (Supplementary Table 2). Furthermore, there was a bias toward use of an IR centromeric to the target IR in the *sae2Δ* strain expressing gRNA-48, whereas most of the inverted duplications from the *sae2Δ rad1Δ* strain had used an alternate IR telomeric to the target IR (*p* = 0.003) (Supplementary Table 2). These data are consistent with a minor role for Rad1–Rad10 nuclease in processing long heterologous flaps.

## Pol32 and Rad51 are required for the formation of inverted duplications

For a hairpin-capped chromosome to form, gap filling DNA synthesis must occur to catch up with extensive resection. Consistent with this idea, Pol32, a non-essential subunit of the DNA Pol δ complex, is required for the formation of hairpin-capped chromosomes in RPA-depleted cells[62]. If the inverted duplications form after foldback

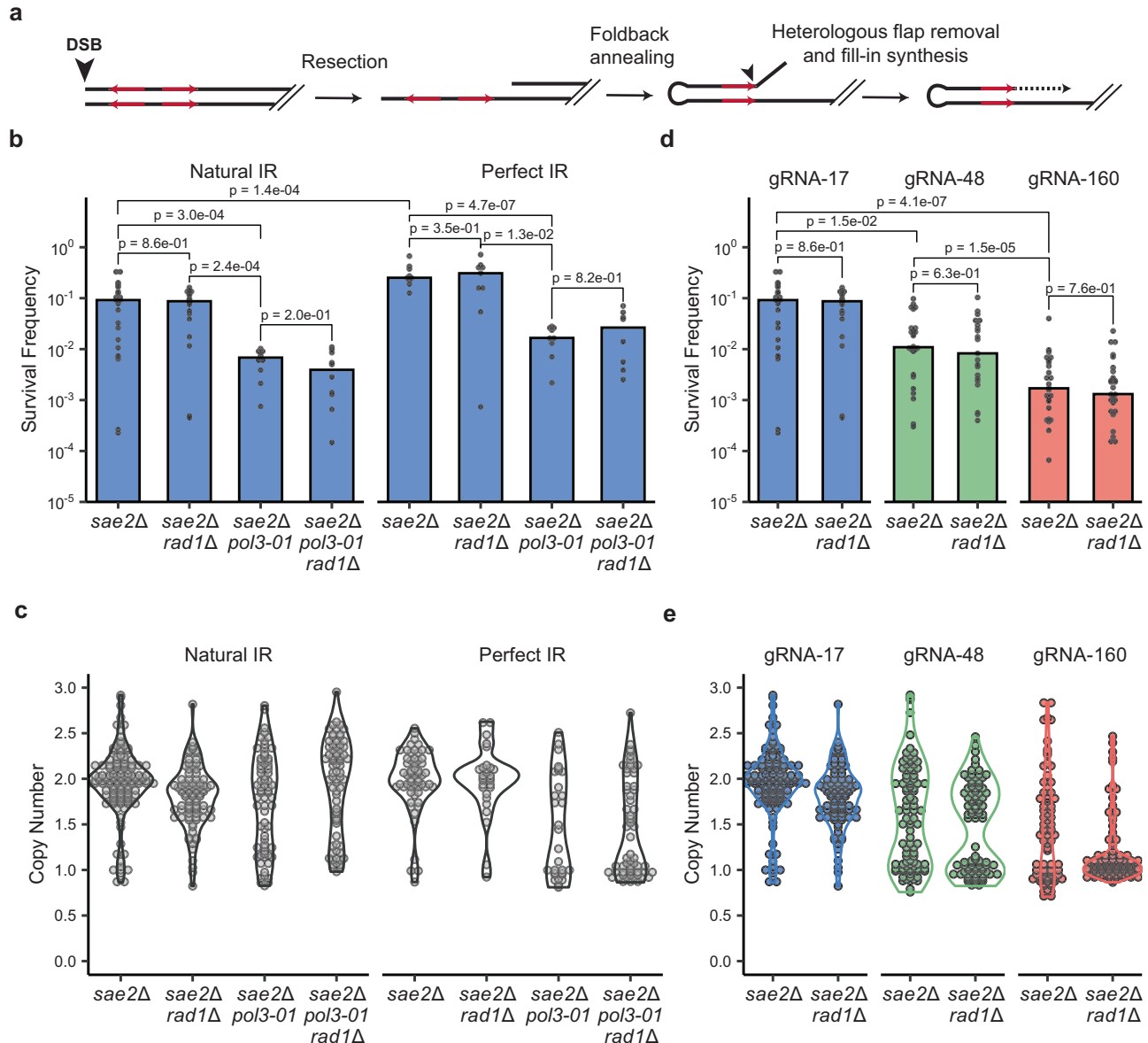

**Fig. 4 | The DNA Pol δ proofreading activity is required for inverted duplication. a** Schematic showing the need for heterologous flap removal prior to fill-in DNA synthesis to form inverted duplications. **b** Survival frequencies of the indicated genotypes with either the natural or perfect IR in response to expression of Cas9 and gRNA-17. *P* values were determined using a two-tailed *t* test. *sae2Δ* natural IR, *n* = 23; *sae2Δ rad1Δ* natural IR, *n* = 18; *sae2Δ pol3-01* natural IR, *n* = 10; *sae2Δ pol3-01 rad1Δ* natural IR, *n* = 10; 4–10 biological replicas. *sae2Δ* perfect IR, *n* = 9; *sae2Δ rad1Δ* perfect IR, *n* = 9; *sae2Δ pol3-01* perfect IR, *n* = 9; *sae2Δ pol3-01 rad1Δ* perfect IR, *n* = 8; 2–3 biological replicas. **c** Independent clones of the indicated genotypes

were analyzed by qPCR to detect increased copy number of sequence between the target IR and centromere. **d** Survival frequencies of *sae2Δ* and *sae2Δ rad1Δ* strains in response to expression of Cas9 and gRNA-17, gRNA-48 or gRNA-160. *P* values were determined using a two-tailed *t* test. *sae2Δ* gRNA-17, *n* = 23; *sae2Δ* gRNA-48, *n* = 26; *sae2Δ* gRNA-160, *n* = 25; *sae2Δ rad1Δ* gRNA-17, *n* = 18; *sae2Δ rad1Δ* gRNA-48, *n* = 21; *sae2Δ rad1Δ* gRNA-160, *n* = 29; 6–10 biological replicas. **e** Independent clones of the indicated genotypes were analyzed as in (**c**). Source data are provided as Source Data Fig. 4.

annealing at the IR, then their formation should require Pol32. As expected, deletion of Pol32 in the *sae2Δ* background significantly decreased the frequency of survival to the Cas9-induced DSB (Fig. 5a), and mostly eliminated the incidence of inverted duplications (Fig. 5b). We found that most of the survivors recovered from *sae2Δ pol32Δ* cells had inverted the sequence between the ER domains within the Cas9 expression cassette, suggesting a strong selection for cells that ablate Cas9 expression. Nonetheless, the reduced survival of *sae2Δ pol32Δ* cells is consistent with a role for Pol32 in formation of inverted duplications. Because Pol32 is also required for BIR[63], the reduced survival of *sae2Δ pol32Δ* cells could also be due to its role in the secondary rearrangement necessary to stabilize a broken dicentric chromosome.

Since most of the inverted duplications described in previous studies are associated with homology-mediated secondary rearrangements[5–7,34], we anticipated that survival would be reduced in *sae2Δ rad51Δ* cells. Although *sae2Δ rad51Δ* cells did indeed exhibit a 28-fold reduction in survival to a DSB as compared to *sae2Δ*, a majority of surviving cells were still able to form inverted duplications (Fig. 5a, b, Supplementary Table 1).

### Secondary rearrangements associated with inverted duplications

To identify the extent of Chr V duplication and secondary recombination events, we analyzed the WGS data from *sae2Δ* inverted

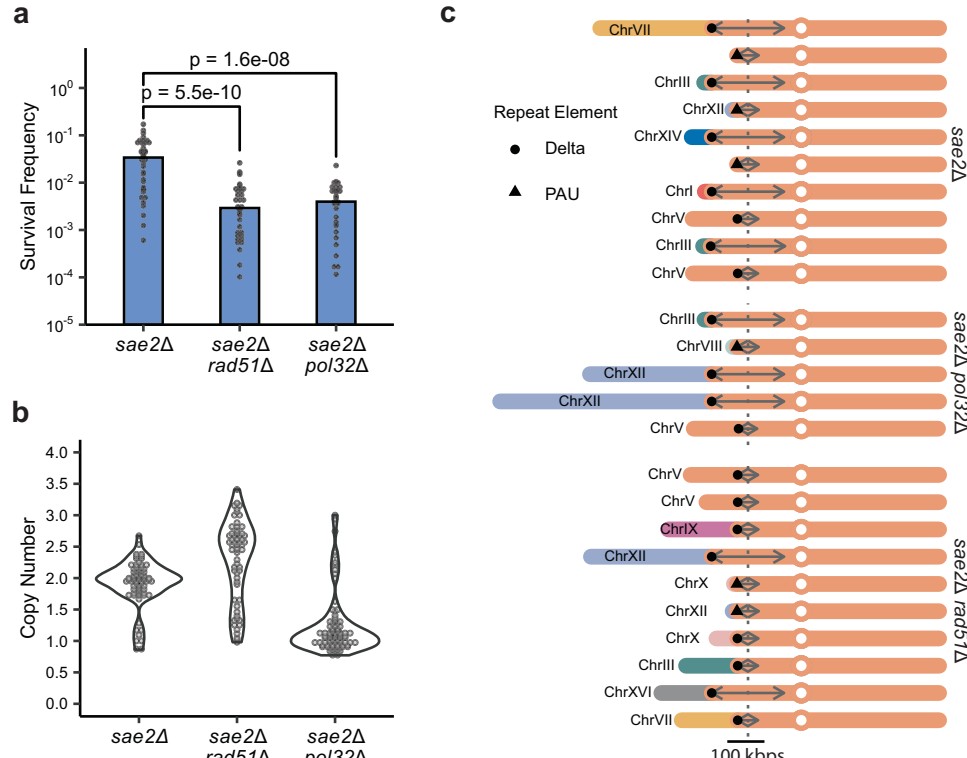

**Fig. 5 | Rad51 and Pol32 contribute to inverted duplication formation. a** Survival frequencies of *sae2Δ*, *sae2Δ rad51Δ* and *sae2Δ pol32Δ*. *P* values were determined using a two-tailed *t* test. *sae2Δ*, *n* = 39; *sae2Δ pol32Δ*, *n* = 32; *sae2Δ rad51Δ*, *n* = 38; 7–11 biological replicas. **b** Independent clones of the indicated genotypes were analyzed by qPCR to detect increased copy between the target IR and centromere. **c** Derivative chromosomes formed by rearrangements following Cas9-induced DSB and inverted duplications as detected by WGS. Connectivity between the chromosomes was determined by analysis of the copy number change, by analysis for structural rearrangements using the Pyrus software suite and by assembly of discordant reads. The dashed vertical line indicates the position of the DSB. Two-headed arrows denote the extent of the inverted duplication. Symbols denote the identity of the repeat sequence at the junction of the rearrangement. Source data are provided as Source Data Fig. 5.

duplication clones to detect copy number variation (CNV) genome wide (Figs. 2d, 5c). Two main regions, at ~62–65 kb and 135–138 kb, demarcate the end point of duplications, coinciding with the locations of delta elements and *PAU2*. *PAU2* is a member of the seripauperin gene family that are ~350 bp in length and show a high degree of DNA sequence homology[64]. There are approximately 21 *PAU* genes in the W303 *S. cerevisiae* genome, mostly located in sub-telomeric regions. In previous studies, the Ty1 element inserted at the *ura3* locus on Chr V-L was identified as a hotspot for secondary rearrangements associated with inverted duplications[5–7]. Notably, the *ura3* locus in the W303 strain background used for this study lacks a Ty1 element and no inverted duplications terminated at this region of Chr V.

To determine the locations of translocations and whether repeat elements mediated them, we examined the structural variations present using Comice[5]. For *sae2Δ*, *sae2Δ rad51Δ* and *sae2Δ pol3-01* inverted duplications, translocations were detected with the breakpoints aligning to either *PAU* genes or delta elements (Fig. 5c, Supplementary Fig. 5a, Table 1). In clones where there is no detectable CNV in a chromosome other than in Chr V, the translocations occurred with sub-telomeric sequences, which are difficult to map. Duplications followed by translocation with sub-telomeric regions (such as those mediated by *PAU* genes) result in derivative chromosome sizes that are not different from the parental Chr V. Therefore, a rearranged chromosome that migrates at the same distance as the parental Chr V by PFGE (Fig. 1d) is not necessarily an indication that an inverted duplication has not occurred. The secondary rearrangements in 5/8 *sae2Δ rad1Δ*, 16/19 *sae2Δ pol3-01*,9/18 *sae2Δ rad1Δ pol3-01* and 5/6 *sae2Δ pol32Δ* inverted duplications also appear to be mediated by delta elements or *PAU* genes (Fig. 5c, Supplementary Fig. 5a, Table 1). In

one *sae2Δ pol3-01 rad1Δ* clone the translocation was mediated by a tRNA.

We also observed some complex rearrangements with evidence for either triplications or quadruplications in 1/11 *sae2Δ*, 2/19 *sae2Δ pol3-01* and 2/8 *sae2Δ rad1Δ* inverted duplication clones (examples are shown in Supplementary Fig. 6), In each case, an inverted repeat marks the junction between a change in copy number. Such events could result from an additional round of foldback priming after breakage of a dicentric chromosome intermediate, as suggested previously[6,7].

Three clones with inverted duplications from WT cells were sequenced and shown to have similar secondary rearrangements to the clones recovered from *sae2Δ* cells. In addition, we sequenced four WT GCR clones that lacked inverted duplications. In two of these, the rearrangements were due to microhomology-mediated non-reciprocal translocations, another non-reciprocal translocation was mediated by 50 bp of sequence homology, and the fourth rearrangement was a Chr V truncation with de novo telomere addition (Supplementary Fig. 5b).

## Rad1 and Pol δ proofreading activity promote homology-mediated secondary events

For one clone each from the *sae2Δ rad1Δ* and *sae2Δ rad1Δ pol3-01* strains, stabilization of the broken dicentric intermediate occurred by de novo telomere addition. Additionally, we recovered 7 clones from the *sae2Δ rad1Δ pol3-01* triple mutant and one clone from *sae2Δ pol32Δ* in which the inverted duplication spans almost the entirety of Chr V minus sequences telomeric to the DSB (two are shown in Fig. 6a, b). PFGE analysis of these clones verified a derivative Chr V that is almost twice the size of the native chromosome (Fig. 6c), indicative of an inversion chromosome. These events initiated at the target IR based

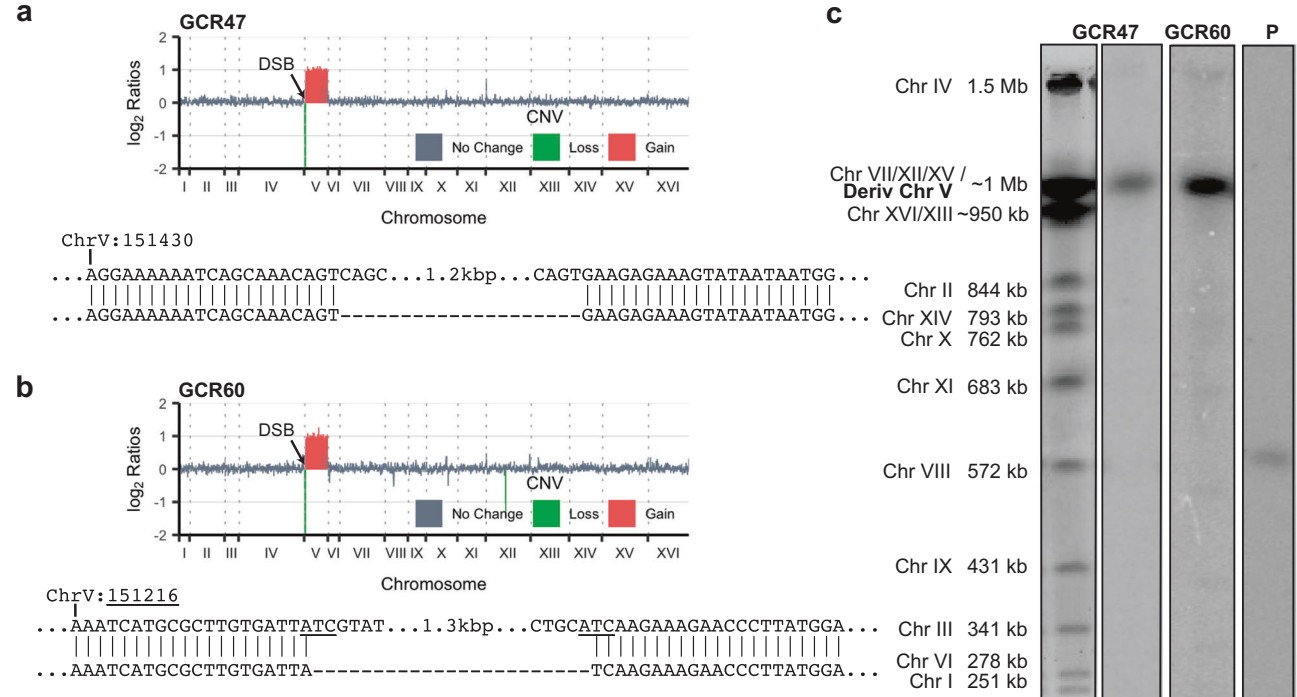

**Fig. 6 | Evidence for a dicentric chromosome precursor. a** and **b** Top: Copy number analysis of two clones that exhibit complete duplication of Chr V centromeric to the target IR. Below: the sequence of junctions spanning a deletion of *CEN5*. **c** PFGE of the same two clones and parental strain followed by Southern blotting using a probe against *PCM1* on Chr V. SYBR Gold-stained PFGE is shown in the left panel to indicate chromosome sizes. Source data are provided as Source Data Fig. 6.

on the recovery of reads spanning the center of the inverted duplication, indicating that they were not products of sister-chromatid fusion. In all cases, we identified a breakpoint junction consistent with an interstitial deletion spanning one of the two copies of the Chr V centromere (Fig. 6a, b, Supplementary Fig. 7). In most cases, the interstitial deletion is limited to ~1 kb on either side of the centromere but can extend for up to 150 kb (Supplementary Fig. 7). Since this type of rearrangement is most prevalent in the *sae2Δ rad1Δ pol3-01* triple mutant, it suggests that homology-directed repair is defective in this genetic background, in which case repair defaults to a mode that leaves evidence of a dicentric precursor. In particular, these data provide strong support for the model that foldback priming near a DSB leads to an inverted duplication that extends from the break to the other end of the chromosome resulting in creation of a dicentric chromosome.

## Discussion

In this work, we show how the DNA sequence context and genetic factors affect the outcome and frequency of DSB-induced inverted duplications. Previous studies have characterized inverted duplications that arise spontaneously[5–7]. However, we find some differences in the genetic requirements and frequencies of inverted duplications when comparing spontaneous with DSB-induced events. This finding suggests that some spontaneous inverted duplications may arise through mechanisms that are different from those induced by DSBs. In agreement with a previous study[7], we find that a major determinant in channeling mutagenic repair is the sequence context of a DSB. In particular, a DSB near an imperfect 11-bp-long IR is a potent inducer of foldback annealing, a precursor to inverted duplications. We found that ~10% of *sae2Δ* and *mre11Δ* cells survived a DSB induced near the target IR, a > 200-fold increase relative to WT cells, of which ~90% were inverted duplications. These data suggest that an initiating DSB is rate-limiting for the formation of inverted duplications, and that Mre11 and

Sae2 play a significant role in resolving hairpin intermediates and preventing foldback inversions.

Inverted duplications recovered from *sae2Δ* cells in spontaneous GCR assays have been shown to contain at their centers short IRs that were derived from the parental sequence[5,6]. This led to the hypothesis that these inverted duplications form by intra-strand foldback annealing of resected DNA at inverted repeats near the site of a DSB (Fig. 7). It was shown that brief induction of a DSB in *sae2Δ* cells increased the rate of GCRs with inverted duplications[7]. In congruence with this study, we found that introduction of a DSB near the target IR was both sufficient and necessary for high frequency formation of inverted duplications in *sae2Δ* cells. Analysis of eleven inverted duplication *sae2Δ* isolates by whole genome sequencing confirmed that the centers of the duplications lie at the targeted IR. The inverted duplications detected in the modified GCR assay mainly used a perfect 15-bp IR, and the deletion of this hotspot did not reduce the spontaneous GCR rate of the *sae2Δ* mutant, but changed the IRs used for inverted duplications[7]. Although we also found use of alternate IRs when the target IR was mutated or a DSB was induced more than 50 bp away from the target IR, the frequency of cell survival was significantly reduced. Furthermore, the fraction of inverted duplications among survivors was decreased and accompanied by an increase in NHEJ repair of the DSB.

The requirement for the IR in the formation of inverted duplications strongly suggests that a foldback at the repeats is an initiating event. Inter-chromosomal annealing between repeats present on two broken sister chromatids is expected to be much less efficient than intra-molecular annealing between nearby sequences. Further support for the foldback model comes from the observation that increasing the distance between the DSB and the IR reduces the survival frequency of *sae2Δ* cells. We imagine that the longer heterologous flap formed following annealing between the repeats results in less stable base-pairing.

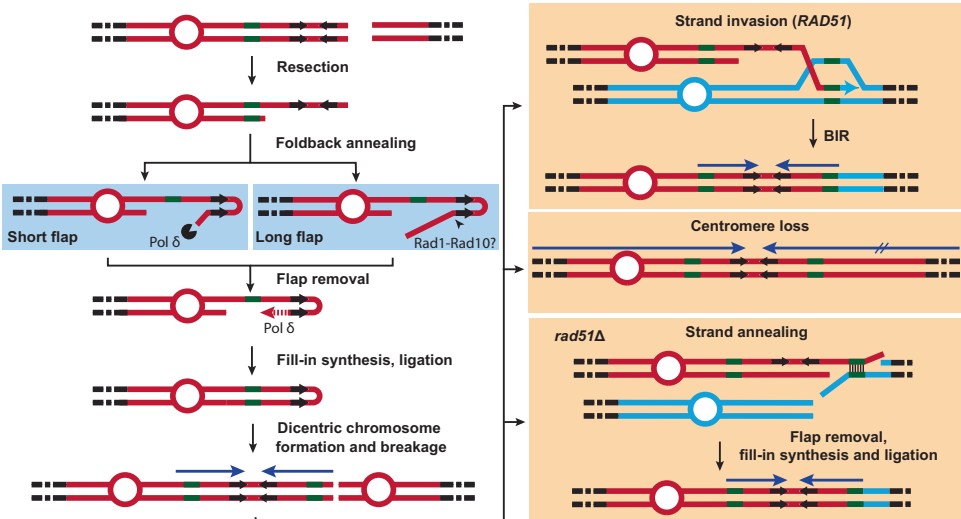

**Fig. 7 | Model for inverted duplication formation.** After DSB end resection, an exposed centromere-proximal inverted repeat (IR) in single-stranded DNA mediates intra-strand foldback annealing. If the IR is near the DSB, foldback annealing forms a short heterologous flap that is removed by the proofreading activity of Pol δ. If the IR is further away from the DSB, a longer flap is formed that may be cleaved by Rad1–Rad10. Following flap removal, Pol δ synthesizes from the 3′-end of the foldback, catching up with the resected end. After ligation, a hairpin-capped chromosome is formed with a terminal deletion. During the next cell cycle, the DNA replication machinery loops around the hairpin end and synthesizes back,

generating a dicentric isochromosome. If the two centromeres of the isochromosome are attached to opposite spindle pole bodies, they are pulled to each nascent daughter cell during mitosis and the chromosome is broken during cytokinesis. The broken chromosome is then stabilized either through the acquisition of a telomere or through centromere loss. The former occurs mainly through BIR initiated by Rad51-mediated strand invasion from a repeat sequence (such as a delta element) near the broken end into a homologous sequence in another chromosome. Alternatively, the broken chromosome can undergo strand annealing with another chromosome through repeat sequences in a Rad51-independent fashion.

A consequence of foldback annealing is the presence of a heterologous flap that needs to be cleaved before fill-in synthesis if the break is at a distance from the IR. Ectopic recombination studies have shown that Rad1–Rad10 is required for cleavage of heterologous flaps that are longer than 20 nt, while the proofreading activity of Pol δ acts redundantly with Rad1–Rad10 to trim shorter flaps[59]. Surprisingly, we found that Rad1 is not required for inverted duplications when the DSB is 20 bp away from the IR, whereas *sae2Δ* cells with a defect in the proofreading activity of Pol δ exhibited a significant defect in the formation of inverted duplications. Furthermore, 9/19 *sae2Δ pol3-01* and 11/18 *sae2Δ rad1Δ pol3-01* of the inverted duplication clones recovered used an IR closer to the DSB, predicted to reduce the size of the heterologous flap to a single nucleotide after foldback annealing. These data suggest that Pol δ proofreading plays a non-redundant role in removing the short heterologous flaps formed. Consistent with our findings, Pol δ proofreading activity is needed to remove 29-nt-long heterologous flaps formed during NHEJ in yeast and trims 2–10 nt flaps generated during Pol θ-mediated end joining in mammalian cells[65,66]. We note that annealing of microhomologies of <14 bp is Rad52 independent, raising the possibility that Rad1 is only required to remove heterologous flaps formed after Rad52-dependent annealing between longer homologies[49,65,67,68]. Although we did not observe a significant decrease in survival of *sae2Δ rad1Δ* cells in response to a DSB induced 51 bp away from the target IR, the fraction of inverted duplications that initiated from the target IR was lower than in *sae2Δ* cells, consistent with a minor role for Rad1–Rad10 in removal of longer heterologous flaps.

The dicentric chromosome predicted to form by replication of the hairpin-capped intermediate would need to be stabilized by loss of one centromere or by acquisition of a telomere following chromosome breakage[26,28]. The recovery of 8 GCRs from *sae2Δ* derivatives with inversion chromosomes and deletion of one centromere provides strong support for this model. A model for inverted duplication formation without hairpin-capped and dicentric chromosome intermediates has been proposed[5,7,22]. In this model, fill-in synthesis

following foldback intra-strand annealing undergoes a template switch at a repeat sequence with a homologous sequence at a different locus, initiating BIR synthesis to the end of the donor chromosome. We cannot rule out that this mechanism is responsible for some inverted duplications described here.

There are multiple mechanisms that can lead to telomere acquisition, including de novo telomere addition or translocation that leads to telomere capture. Translocations have been shown to mainly occur by HR using repeat elements[5–7,27]. In agreement with previous studies[5–7], the majority of secondary rearrangements we observed involved repeat sequences. We observed only two instances of de novo telomere addition suggesting that this mode of repair is rare. We detected rearrangements that mainly used delta elements and *PAU* repeats. Translocations involving *PAU2* have also been detected in spontaneous inverted duplications[7,34,54]. Previous work showed breakage of dicentric chromosomes occurs within a 25–30 kb window of the centromere[24,25]. The cluster of delta elements located ~14 kb away from the centromere (*YELCdelta4, YELWdelta5* and *YELWdelta6*) falls in the middle range of the breakage window. Consistently, 23/55 inverted duplications associated with secondary recombination events used this cluster of delta elements. The next delta element, *YELCdelta1/2*, is 50 kb from the centromere but is shorter than other LTRs (174 bp vs ~300–360 bp) and was used in 17 inverted duplications. *PAU2*, which is located ~30 kb centromeric to the DSB and ~90 kb from the centromere, was used in 14 inverted duplications. Since more than half of the inverted duplications sequenced used *YELCdelta1/2* or *PAU2*, the majority of breakage events likely occur further telomeric from *YELWdelta4-6* cluster. The bias for use of delta elements is likely due to the high copy number of these sequences providing many potential templates for repair.

Since breakage of the dicentric chromosome is unlikely to occur within the repeats used for secondary events, Rad51-dependent strand invasion would require removal of heterologous flaps. Given the role of Rad1 in heterologous flap cleavage during HR and SSA[57], the high survival frequency of *sae2Δ rad1Δ* and use of a repeat sequence for the

secondary events is surprising. However, two of the inverted duplications recovered from the *sae2Δ rad1Δ* mutant had complex rearrangements and one was associated with telomere addition, suggesting that stabilization of a broken dicentric intermediate by HR might be more problematic in cells lacking Rad1–Rad10 nuclease. Moreover, 7 clones from the *sae2Δ rad1Δ pol3-01* mutant had inversion chromosomes with a centromere deletion and one was associated with de novo telomere addition. This finding suggests that Rad1 and Pol δ proofreading activity may act redundantly to remove heterologous flaps during strand invasion between repeat sequences.

The long homology involved in the secondary rearrangements and non-reciprocal nature of the rearrangements point to a BIR mechanism or a crossover in which only one product is recovered in a daughter cell. Further support for either mechanism is the significant reduction in survival frequencies of *sae2Δ rad51Δ* and *sae2Δ pol32Δ* mutants compared to *sae2Δ*. This finding indicates that the secondary rearrangements generally involve a strand invasion step. The survivors with inverted duplications detected in *sae2Δ rad51Δ* cells are associated with homology-mediated secondary events, potentially arising by SSA involving another broken chromosome (Fig. 7). The increase in spontaneous lesions in *rad51Δ* cells may allow for the detection of rearrangements using these other mechanisms by increasing the likelihood of the presence of a second broken chromosome[69,70]. Interestingly, we recovered fewer inverted duplications in *sae2Δ pol32Δ* cells than in *sae2Δ rad51Δ* cells, suggesting an additional role for Pol32 in formation of inverted duplications. Since a previous study reported fewer DSB-induced hairpin-capped chromosomes in RPA-depleted *pol32Δ* cells[62], and Pol32 has also been implicated in microhomology-mediated end joining[67,71], this function is likely to be at the fill-in synthesis step after foldback annealing and heterologous flap trimming.

Foldback inversions have been detected in a variety of human cancers, including pancreatic, ovarian, breast, and squamous cell carcinomas (SCC), and are associated with poor prognosis[72–78]. The mechanism by which these foldback inversions form is unknown and could involve the foldback priming mechanism described here, particularly in tumors with mutations in *MRE11, RAD50* or *NBS1*. However, we note that human cells have more active end joining mechanisms than yeast[79]; therefore, fusion of broken sister chromatids is likely to contribute significantly to the generation of foldback inversions.

## Methods
### Strains, media, and growth conditions
Yeast strains used in this study are derived from W303 and are listed in Supplementary Table 3. All yeast strains and plasmids are available by written request to the corresponding author. Yeast strains were made either by genetic crosses or transformation. Transformation was performed using the LiAc/ssDNA carrier DNA/PEG method[80]. Cells were grown in either yeast rich media (YPD) or synthetic complete (SC) media with amino acid dropouts as described previously[81]. β-estradiol-containing plates were made by spreading β-estradiol on YPD plates to 5 μM before use. For GAL-HO assays, cells were pre-grown in YP medium with 2% raffinose and plated on YP medium with 2% galactose for HO induction. All cells were propagated at 30 °C. pAA3, pAA13, pAA16, pAA18, pAA19, pAA20, pAA21 and pAA22 were integrated into the yeast genome by digesting the plasmids with AscI and transforming cells with the linearized product. Integration is confirmed by multiplex PCR as described previously[82].

Complete deletion of the *RAD1, RAD51*, and *POL32* ORFs was achieved by one-step gene disruption. For each, a PCR fragment was made by amplifying either HphMX (*rad1Δ*) or NatMX (*rad51Δ* and *pol32Δ*) from pAG32 (Addgene plasmid # 35122) or pAG25 (Addgene plasmid # 35121), respectively[83]. Forward (TCGACGGATCCCCGGGT-TAA) and reverse (AATTCGAGCTCGTTTTCGACACT) primers contained 60 bp of sequences immediately upstream and downstream of

the ORFs, respectively. Yeast cells were then transformed with the fragment, plated on YPD and then replica plated on the appropriate selection media after one day of growth. Gene deletion was confirmed by multiplex PCR.

The inverted repeat was mutated by CRISPR/Cas9 gene editing as described previously[84]. The gRNA IR_gRNAmut2 was cloned into pCeASY and used for editing (Supplementary Table 4). The repair template included the desired mutations along with 500 bp of homologies telomeric (amplified using IR_500_upstream_F and mutation-specific reverse primer) and centromeric to the target site (amplified using IR_500_downstream_R and a mutation-specific forward primer) and was assembled by overlap extension PCR using Phusion polymerase. The targeted modifications were confirmed by sequencing.

The *pol3-01* allele was made by CRISPR/Cas9 gene editing as described above. The repair template containing the *pol3-01* mutation[60] along with ~75 bp of upstream and downstream homology sequences was assembled by PCR using the primers pol3-01_middle, pol3-01_left and pol3-01_right, listed in Table S4. The mutation was confirmed by sequencing of DNA amplified by pol3-01_mut_F and pol3-01_mut_R.

The gRNA-17 cleavage site was replaced by an HO cleavage site by CRISPR/Cas9 gene editing using the gRNA_mut1 targeting sequence and oligos listed in Supplementary Table 4. The repair template contained a 36-bp-long HO recognition and 500 bp of homologies telomeric and centromeric to the gRNA target site, respectively. The telomeric homology was amplified from the genome using IR_500_upstream_F and HOcs_gRNA2_mut_R3, and the 500 bp centromeric homology was amplified from the genome using IR_500_downstream_R and HOcs_gRNA2_mut_F3. The two fragments were then assembled by PCR using the oligos IR_500_upstream_F and IR_500_downstream_R. The mutation was confirmed by DNA sequencing.

### Plasmids and constructs
The oligonucleotides used for gRNAs and genome editing are listed in Supplementary Table 4, and plasmids are listed in Supplementary Table 5. Details of the oligonucleotides used for plasmid modifications and assembly are available on request from the corresponding author.

Cas9 expression plasmids: pAA1 was made by cloning the *GAL1* promoter into pML107 (Addgene plasmid # 67639)[85]. The *GAL1p* promoter was amplified by PCR from genomic DNA and assembled into NcoI and SpeI digested pML107 using the HiFi DNA Assembly mix (NEB # E5520). The gRNA scaffold sequence was further modified to replace the SwaI and BclI fragment with a BaeI fragment from pML107 by HiFi Assembly. pAA3 was made by cloning $P_{GAL1}$-CAS9-NLS-6xGLY-FLAG-$T_{CYC1}$ into pRG203MX[82]. $P_{GAL1}$-CAS9-NLS was amplified from pAA1 and the *CYC1* terminator was amplified from genomic DNA. The two fragments were then assembled into SpeI- and EcoRV-digested pRG203MX. The lexO expression system is a two-module system composed of the promoter and the transcription factor. The promoter is composed of four tandem repeats of the lexA box fused to the minimal *CYC1* promoter (together referred to as $P_{lexO}$)[39]. The transcription factor is LexA gene fused to an estrogen receptor and a B112 transcriptional activator under the control of *ACT1* promoter (together referred to as $P_{ACT1}$-LexA-ER-AD)[39]. The two modules were cloned into a single plasmid to generate pAA12, which was then used for subsequent cloning. pAA16 was made by amplifying *CAS9-NLS-FLAG* from pAA3 and was cloned by HiFi assembly into ApaI-SacI-digested pAA12. pAA18 was made by HiFi assembly: *CAS9* (amplified from pAA3), ER-LBD (amplified from pRG634) and NLS-FLAG (amplified from pAA3) were assembled with ApaI-SacI-digested pAA12. The orientations of $P_{lexO}$-CAS9-NLS-FLAG-$T_{ADH1}$ and $P_{lexO}$-CAS9-NLS-FLAG-ER-$T_{ADH1}$ were reversed relative to $P_{ACT1}$-LexA-ER-B112-$T_{CYC1}$ by re-cloning the NotI fragment of pAA19 and pAA20 respectively, into pAA12.

Integrating gRNA constructs: A version of pCAS (Addgene plasmid # 60847)[84] in which the gRNA cloning sequences were replaced with an XbaI and ZraI fragment was kindly provided by R. Gnügge. The sgRNA from this plasmid, including the tRNA promoter, the HDV ribosome, the sgRNA sequence and the sNR52 terminator, was PCR amplified and cloned into ApaI- and XhoI-digested pRG205MX[82] whose backbone XbaI and ZraI sites were then destroyed by site-directed mutagenesis. This vector was named pAA9. The gRNA targeting sequences were cloned into pAA9 by annealing oligos pCeASY-gRNA-S and pCeASY-gRNA-AS (where 20 × N in Table S4 represent the sense and anti-sense gRNA target, respectively) and ligating the annealing product to ZraI- and XbaI-digested pAA9.

Cas9-mediated gene editing plasmids: pCeASY, a version of pCAS (Addgene plasmid # 60847)[84] in which the gRNA cloning sequences were replaced with an XbaI and ZraI fragment was kindly provided by R. Gnügge. gRNA targeting sequences were cloned into this plasmid as described above.

## Survival assays

All survival assays with *lexO-CAS9-ER* were done in strains in which the relevant gRNA was integrated into the genome. Fresh single colonies from each strain were grown overnight in YPD, diluted in the morning and grown to early log phase. Cells were then diluted and plated on YPD ± β-estradiol. For the strains with the HOcs replacing the gRNA-17 binding site, HO endonuclease was expressed from the *GAL* promoter. Fresh single colonies from each strain were grown overnight in YPD, diluted into YPR in the morning and grown to early log phase. Cells were then diluted and plated on YPD or YPGal plates. The survival frequency was measured by:

$$\frac{(\#\text{colonies on YPD} + \beta - \text{estradiol (or YPGal) plates} \times \text{fold dilution})}{(\#\text{colonies on YPD plates} \times \text{fold dilution})}$$

$$(1)$$

## Statistics and Reproducibility

Data from survival assays represent independent cultures (*n*) from at least two biological replicates as indicated in the figure legends. *P* values for survival assays were determined using a two-tailed *t* test. Differences in IR usage between *sae2Δ* and *sae2Δ rad1Δ* survivors were determined by Fisher's exact test.

## Screening of survivors for mutagenic repair

Colonies that formed within two days of plating on β-estradiol-containing medium had inverted the expression cassette eliminating Cas9 expression and were excluded from colony counts. These events were rare for most strains. PCR and qPCR screens were done using DNA extracted from boiled cells. Colonies were picked from the survival assay plates, spread onto patches on YPD plates and grown overnight. A colony-sized amount of cells was then boiled in 50 μl 0.2% SDS at 95 °C for 5 min. All screening PCR reactions were performed using DreamTaq (ThermoFisher # EP0711) in the presence of 1% Triton X-100 and using 1 μl of boiled cell lysates in a 20 μl reaction. Control primers MEC1-F2 and MEC1-R2 were used in the same PCR tubes for all reactions (Supplementary Table 4). The PCR screen to detect NHEJ products was done using primers P3 and P4. For screening for retention of the terminal part of the left arm of chromosome V, the same PCR reaction was used but with primers P1 and P2 instead of the P3/P4 primers. WT clones for which there was no P3/P4 but which gave P1/P2 product were further screened using primer set P7 and P8 to detect homeologous gene conversion events that lead to loss of P4 priming sequences. Primer sets P11/P12 and P13/P14 were used to screen survivors from gRNA-48 and gRNA-160 expression for use of the target IR. Clones that were negative for P11/P12 and positive for P13/14 were scored as target IR inverted duplications, clones that were positive for

both diagnostic PCRs were scored as using an IR telomeric to the target IR, and clones that were negative for both PCRs were scored as using an IR centromeric to the target IR. We identified a sub-population of fast-growing survivors from WT and *sae2Δ pol32Δ* cells that had no alteration at the gRNA cut site. These were shown by a PCR assay to have inverted the DNA segment between the ER domains in the expression construct resulting in failure to express Cas9.

To screen for inverted duplications, DNA was prepared from cells that were serially passaged for a total of two times (from survival plate to "patch 1," then from "patch 1" to "patch 2"). Boiled DNA (as above) was diluted 10-fold in 2% Tween 20, of which 4.4 μl was used in a 10 μl SYBR Green qPCR reaction. Primer set ChrV-60K-F and ChrV-60K-R (P5 and P6) were used to probe for the duplication on Chr V, and ADH1-fwd and AHD1-rev were used for a reference amplicon. Each reaction was run in triplicates. During each qPCR run, two parental controls were run to normalize the sample Cq values to. The copy number was measured by: $\frac{(E_{ChrV60K})^{\Delta C_q(P-S)}}{(E_{ADH1})^{\Delta C_q(P-S)}}$ (2), were $E_{ChrV60K}$ and $E_{ADH1}$ are the primer efficiencies for the *ChrV-60K* and *ADH1* amplicons, respectively, and $\Delta C_q(P-S)$ is the difference between the quantification cycles of the parental and surviving clones.

## Pulse-field gel electrophoresis (PFGE)

DNA for PFGE was extracted in low-melting point agarose from cells grown to saturation as described previously[86]. The chromosomes were separated in a BioRad CHEF-DR II. The gel was then stained using SYBR Gold (Invitrogen # S11494) for 1 h. The chromosomes from the gel were transferred to nylon membranes (Hybond N+) and hybridized to radioactively labeled *PCM1* to identify Chr V fragments.

## Southern blotting to detect inverted duplications

A total of 3–5 μg of DNA isolated from surviving clones was digested for 8 h with 15 units of the indicated restriction enzymes, separated on agarose gels and transferred to nylon membranes (Hybond N+). The DNA was then hybridized with breakpoint-specific radiolabeled probe to detect inverted duplications.

## Whole genome sequencing

Genomic DNA for library preparation was isolated using YeaStar Genomic DNA Kit (Zymo Research # D2002). DNA libraries were prepared using the Illumina DNA Library Prep kit (Illumina # 20018704). For each sample 100 ng of genomic DNA was used following the manufacturer's instructions with the following exceptions: we used half the recommended volume of each reagent, and we modified the size selection scheme to achieve larger fragments as follows: amplified tagmented DNA was diluted 1:4.44 in water (22.5uL DNA in 77.5uL water). During the first size selection step, 44.7uL SPB was added to diluted DNA. After mixing and incubation, 142 μl of the supernatant was used during the second size selection step with 10 μl SPB. Each library was resuspended in 30 μl buffer and the libraries were then pooled in equimolar amounts. After pooling, a final 0.4 × SPB size selection (40 μl SPB added to 50 μl pool + 50 μl water) was performed in order to eliminate small fragments prior to submitting the pool for sequencing. Pooled libraries were diluted and denatured according to the manufacturer's recommendations. Paired-end read sequencing was done in a NextSeq 500/550 platform using either the 150-cycle Medium Output Kit (Illumina # 20024909), 75 cycles per read, or the 75 cycle High Output Kit (Illumina # 20024906), 37 cycles per read.

All mapping was done to a W303 reference genome (PRJNA324291)[87]. For all analysis, all reads were quality filtered with fastp using default settings before mapping[88]. For copy number variation analysis (CNV), the paired-end reads were mapped using Bowtie2[89]. Samtools[90] was used to remove PCR duplicates and to extract sequencing depths for each position along the genome for each sample. CNV was calculated relative to a parental clone that was

sequenced with the GCR samples. Briefly, for each the parental and the GCR clone, the number of reads mapping to each position was normalized to the total number of reads. Next, the chromosome positions were binned (either in 5 kb or 1 kb windows, depending on the analyses shown in the figures). Finally, the relative copy number was calculated for each bin by taking the ratio of the normalized number of reads for each bin from the GCR clone to that of the parental clone. A constant of 0.0625 was added to each computed CNV value to allow for $\log_2$ transformation. For significant value calculation, first, the same CNV analysis was done by randomly splitting the parental reads into two and finding the CNV between the split samples. Then, the CNV values for the split parental samples were used as the null distribution, which was then used to determine the $p$ value of the CNV for each bin of the GCR sample. Finally, multiple testing correction was done using the false discovery rate method. The threshold for significance was set at 0.001. Code used for CNV analysis is provided in the Supplementary Software file.

Structural variation (SV) was detected using Comice, part of the Pyrus suite[5]. The reads for each sample were aligned separately using Bowtie2 and then processed using Comice. The sequence at the center of the SVs was obtained using two strategies: the first was from the outcome of Comice, the second was form de novo assembly of the discordant reads near the boundary of the SV variants. For the second strategy, SV boundaries were determined based on the CNV data. Next, discordant reads pairs for which one mate pair maps to within a few kb of the SV boundary were extracted from the mapped BAM files using samtools. The extracted reads were then de novo assembled using Unicycler[91].

### Reporting summary

Further information on research design is available in the Nature Portfolio Reporting Summary linked to this article.

## Data availability

The cell survival and qPCR data are provided in the Source Data files. WGS data discussed in this publication has been deposited in the Sequence Read Archive (SRA) database and are accessible through accession code PRJNA900608. Source data are provided with this paper.

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

## Acknowledgements

We thank C. Putnam for advice on WGS analysis and members of the Symington lab for review of the manuscript. This work was supported by grants from the National Institutes of Health (P01 CA174653 and R35 GM126997 to L.S.S.), and in part through the NIH/NCI Cancer Center Support Grant P30CA013696 (CCSG DNA Sequencing Core).

## Author contributions

A.M.A. performed most of the experiments shown in Figs. 1–6 and Supplementary Figs. 1–7; M.R.N. contributed to the experiments in Figs. 1, 4–6 and Supplementary Figs. 3–7; and I.A. contributed to Supplementary Table 1. A.M.A., M.R.N., and L.S.S. contributed to the study design, data analysis and manuscript preparation.

## Competing interests

The authors declare no competing interests.
