## [Peer Review File · Nature Communications]

Double-strand breaks induce inverted duplication
chromosome rearrangements by a DNA polymerase
 δ -dependent mechanismREVIEWER COMMENTS

Reviewer #1 (Remarks to the Author):

This elegant study addresses an important and underexplored source of genome instability where DNA breaks near short inverted repeats (IR) lead to dicentric/acentric isochromosomes. Using a sophisticated yeast genetic approach, the authors convincingly show that this pathway is significant at native IR in the absence of Mre11 endonuclease activity. They also demonstrate that the following sequence of events leads to isochromosomes: (i) double-strand break, (ii) 5' resection, (iii) ssDNA IR fold-back, (iv) removal of the heterologous tail by Pol delta proofreading activity with a minor contribution of the Rad1/Rad10 flap endonuclease, (v) fill-in synthesis by Pol delta and (vi) replication of the hairpin-capped chromosome to generate an isochromosome. Mre11 endonuclease activity likely prevents this pathway by opening the hairpin end prior to its replication. This work also sheds new light on the respective contribution of Sae2/Mre11, Pol32 and Rad1/10 for genome stability.

The experiments are beautifully designed and their interpretation quite rigorous. It is a real pleasure to read the manuscript.

Cleavage of the hairpin by Sae2/Mre11 is the most likely scenario, but can the authors rule out a contribution of 5' end resection? I have in mind a previous study from David Lydall where Exo1 loss promotes IR-induced isochromosomes (DOI: 10.1101/gad.316504). What would be the impact of Exo1 loss in the new assay?

Minor point:

In Figure 2A, the expected BamHI fragments at 1.2 and 11.2 kb are missing. Is there an explanation?

Reviewer #2 (Remarks to the Author):

Previous studies have found that MRE11 and SAE2 suppress the formation of inverted duplications in budding yeast most likely by the cleavage of DNA hairpins formed from single-stranded inverted DNA sequences that are exposed following the resection of an adjacent DSB. This new paper from the Symington lab provides additional support for this model and highlights a role for the proofreading activity of Pol δ in stabilising the putative ssDNA hairpin by processing DNA flaps, as well as a potential role for Pol32 in converting the hairpin into a full-fledged hairpin capped chromosome. Prior studies have proposed that hairpin capped chromosomes are replicated to form dicentric isochromosomes that are either stabilized by the loss of one centromere or subsequently break during cell division leading to chromosome stabilization by a secondary DNA rearrangement. Consistent with these models, Al-Zain and colleagues identify both types of chromosomes amongst clones with inverted duplications induced by CRISPR/Cas9 DSBs.

Overall, this a nicely presented study that provides additional support for existing concepts and models that have been promoted and explored in other recent studies (e.g. Deng et al 2015; Li et al 2020).

Comments and questions:

1. It has been shown that Cas9 releases very slowly from DNA post-cleavage and therefore can inhibit subsequent DNA repair. It has also been shown that this stable post-cleavage complex can release the 3' end of the cleaved non-target strand, which can then be degraded by nucleases. Based on the information in Figure 1A, it would appear that the 3' end of the cleaved non-target strand is on the centromere proximal side of the DSB. Degradation of this end (possibly by Mre11) could lead to a foldback intermediate with a 5' heterologous DNA flap. This is just one example of how the specific

properties of CRISPR/Cas9 DSBs could influence the subsequent DNA repair pathways and impinge on data interpretation. It would be prudent to determine to what extent, if any, the reported repair outcomes are influenced by the properties of the CRISPR/Cas9 DSB. As the authors point out in their Discussion (page 12), there are differences in the genetic requirements and frequencies of inverted duplications when comparing spontaneous with their DSB-induced events. How much of this difference is due to the nature of the CRISPR/Cas9 DSB? The authors should consider testing whether the gRNA target and non-target strands have any impact on the genetic dependencies of inversion deletion formation (i.e. whether the released 3' end is on the telomere or centromere side of the DSB), and whether an I-SceI DSB generates similar outcomes.

2. I'm intrigued by the multiple Chromosome V bands detected in Figure 1D, which are particularly prevalent in the *sae2Δ* mutant. The authors attribute these bands to a heterogenous population of cells that have undergone different rearrangements. Does this mean that the efficiency of DSB formation varies from cell to cell (i.e. not all cells experience a DSB before they divide) and/or that DSBs can be repaired faithfully so that the DNA can be subject to further rounds of cleavage? If so, then how do we know that the efficiency of DSB formation and/or faithful repair doesn't vary from strain to strain and affect survival frequency and the chances of detecting a particular GCR outcome? I also find it surprising that wild-type clone 2 and *sae2Δ* clone 7 (Figure 1D) have what appears to be an identical pattern of chromosome V bands. What are the chances that the same heterogenous population of cells would occur in two different clonal populations from two different strains in such a small sample size?

3. The authors mention that inversion between ER domains can eliminate Cas9 expression. These inversions account for ~20% of colonies growing on inducing conditions for wild-type cells. Were other strains checked for the occurrence of this rearrangement to exclude the possibility that this was contributing to survival frequency?

4. Most of the Figures and figure legends do not include the sample size (n).

Reviewer #3 (Remarks to the Author):

In this study the authors investigate the effect of induced double strand breaks (DSBs) on the formation of Gross Chromosomal Rearrangements (GCRs). They observed that a DSB induced near a short inverted repeat results in the formation of inversion GCRs mediated by foldback inversions mediated by the short inverted repeat and that these GCRs are suppressed by Sae2 and the Mre11 complex. They provide extensive genetic analysis of this type of GCR formation along with detailed structural characterization using PCR, PFGE, qPCR and whole genome sequencing. Some of these methods are novel. And the analysis, such as mutagenesis of the initiating inverted repeat, is very thorough. In aggregate, the results provide a great deal of information about the mechanisms underlying the formation of inversion GCRs; these results, obtained in yeast, are of considerable relevance to the formation of these types of GCRs in cancer.

Overall, this is a very well-done study that is very thorough and uses a diversity of complementary, sometimes new, methods to generate data that support the conclusions presented. The paper is interesting and well written. Some questions about the experiments, data presentation and interpretations and conclusions are noted below, but overall, the conclusions seem solid. The results presented extend what we know about the formation of a particularly interesting class of GCRs, the inverted repeat mediated inversion GCRs. The most novel aspect of the study is the genetic analysis of DSB induced inversion GCRs which is in many ways the main focus of the paper; other than the analysis of Sae2, this analysis goes well beyond that of previous studies and is interesting and insightful.

Specific comments

Abstract. I think the abstract over-states some key results such as the supposed strong requirement in inversion formation such as the requirement of Poldelta Exo activity, POL32, RAD1 and RAD51 in different steps in GCR formation, which as note below are partial and sometimes more subtle than an absolute requirement. It also implies that less was previously known about inversion formation, even in the case of DSB induced events, than is the case. Regardless, the paper does describe new results. The abstract needs to be rewritten to provide a better description of the results present and their context.

P2-4. The introduction section is excellent. Informative and scholarly.

P4, L1. Ref 7 also carried out studies on this point so this is not correct. But obviously, this issue is a broad one worthy of multiple studies using different assays and target chromosomes, like the studies presented in the present paper.

Fig 1 and others. A concern I have is how accurate a measurement is survival for the rate or efficiency of GCR formation. Related to this is what are adequate 100% and 0% GCR formation controls and how should relative survival data be presented. Absolute survival, as presented is useful. But also should survival relative to WT survival or relative to sae2 mutant survival be presented? This is particularly true given that the different mutations studied could affect survival in ways unrelated to the rate of GCR formation. I don't doubt the conclusions, but possibly the limitations should be explained. I find the changes in the spectrum of GCRs formed under different experimental conditions to be useful and supportive of the authors conclusions; in this regard, the authors miss some opportunities to explain how GCR spectrum data support the conclusions from the survival data.

P6, L6-15 and Fig S2. This result would be easier to understand if Fig S2 better indicated with better precision where the P3/4 and P7/8 primers map. For example, could the primer sites be indicated on the CAN1 sequence in Fig S2D or a modified version. This would help explain the presence of P7/8 products in isolates that did not yield P3/4 products.

P6, L21-23. There are some references that could be cited here regarding the formation of these types of rearrangements. The same is true for expanded chromosome sized rearrangements seen by PFG.

P7, L2. Had likely repaired the DSB by NHEJ. Fig 2A does not provide enough information on the location of the Bam, EcoRV and Pst sites on the telomeric side of the DSB to allow drawing any mapping conclusions. I'm sure the authors could add this information.

P7, L26. Is this what the authors define as a long spacer in Table 1? The legend to Table 1 could be improved.

P8, L21-32. The results show that the pol3-01 mutation reduces survival of the sae2 mutant but not to the level seen in wild-type and that a significant fraction of the survivors in the sae2 pol3-01 mutant appear to be inversions. Shouldn't it be more clearly stated that this is consistent with a partial defect in flap processing? Given the partial defect caused by pol3-01, did the authors examine other nucleases?

P9, L1-6. Did the authors sequences any of the inversions seen in the sae2 pol3-01 mutant as it appears that these can be recovered? This would test whether the pol3-01 mutation alters the types of inversions seen. This is certainly what the comparison of sae2 rad1 with sae2 rad1 pol3-01 shows. This could be more clearly stated.

P9, L23 to P10, L7. How does the effect of changing the DSB site compare to the results described in Ref 7 where different DSB sites were examined. Further, in regard to the conclusion about RAD1,

didn't Ref 7 conclude that defects in Rad1-Rad10 caused little of no change in inversion frequency but did cause changes in the inversion sites used and thus played some role in flap processing and inversion formation.

P10, L8-18 and Table 1. Having provided convincing evidence for the formation of several dicentric chromosomes. I'm not sure I'd title this section "evidence for a dicentric chromosome intermediate" unless there is convincing evidence that yet other rearrangements involved dicentric intermediates. Also, having called them dicentric, why then call them isochromosomes, especially in Table 1. Would "inversion chromosomes" or "dicentric inversion chromosomes" be better?

P10, L18 to P11, L11. I find this confusing. The authors are trying to argue that inversions in a *sae2* mutant require POL32 and RAD51 based on reduced survival in response to a DSB. There is reduced survival but not to WT survival levels. This is consistent with a partial requirement for POL32 and RAD51. Indeed, consistent with this, Fig 6E shows that inversions are formed in each case, obviously at different rates. I think the authors are pushing a view that POL32 is absolutely required when in fact the *pol32* defect is partial and more subtle in that the length of the inverted region is shorter, an interesting result that should be highlighted rather than obscured. This result is not as different from that of Ref 7 as the authors imply.

P10, L26. I don't agree with this statement. What is reduced is duplication of the qPCR assay site, which is quite a bit centromeric from the inversion initiation site. Fig 6E and Table 1 actually show that the virtually all of the products recovered are inversions, albeit ones that duplicate a shorter region and don't duplicate the qPCR assay site. Again, an interesting result that is obscured by the desire to claim that POL32 is absolutely required.

The qPCR assay is an interesting innovation in these studies. Related to the above point, the authors should point out the limitations of the assay related to the choice of locations for the qPCR primer sites as this is relevant to the interpretation of the data.

P12, L7. A dicentric intermediate is not necessarily involved. For example, strand switching during the inversion primed DNA synthesis could also account for the secondary rearrangements. All of the possibilities should be mentioned. Also, Ref 24 is relevant here.

P12, L17. Ref 7 also extensively studied DSB induced inversions.

P12, L21. Another explanation is that different substrates were used in the different studies, and that this effects genetic outcomes.

P12, L21-24. Wasn't this proposed in Ref 7.

P14, L6-15. It should be noted that Ref 7 also extensively presented models involving dicentric chromosomes.

Could the Discussion be compacted a bit to focus on the key points.

We thank the Reviewers for their positive comments and suggestions for additional experiments to strengthen the conclusions of our study. Below, we provide a point-by-point response (in blue font) to the concerns raised by the Reviewers:

Reviewer #1

This elegant study addresses an important and underexplored source of genome instability where DNA breaks near short inverted repeats (IR) lead to dicentric/acentric isochromosomes. Using a sophisticated yeast genetic approach, the authors convincingly show that this pathway is significant at native IR in the absence of Mre11 endonuclease activity. They also demonstrate that the following sequence of events leads to isochromosomes: (i) double-strand break, (ii) 5' resection, (iii) ssDNA IR fold-back, (iv) removal of the heterologous tail by Pol delta proofreading activity with a minor contribution of the Rad1/Rad10 flap endonuclease, (v) fill-in synthesis by Pol delta and (vi) replication of the hairpin-capped chromosome to generate an isochromosome. Mre11 endonuclease activity likely prevents this pathway by opening the hairpin end prior to its replication. This work also sheds new light on the respective contribution of Sae2/Mre11, Pol32 and Rad1/10 for genome stability.

The experiments are beautifully designed and their interpretation quite rigorous. It is a real pleasure to read the manuscript.

Cleavage of the hairpin by Sae2/Mre11 is the most likely scenario, but can the authors rule out a contribution of 5' end resection? I have in mind a previous study from David Lydall where Exo1 loss promotes IR-induced isochromosomes (DOI: 10.1101/gad.316504). What would be the impact of Exo1 loss in the new assay?

Thank you for alerting us to the Lydall study showing that loss of Exo1 rescues the lethality of telomerase and Rad52-defective cells by a mechanism involving formation of isochromosomes initiated at short, inverted repeats. This work is now cited in the Introduction.

In a previous study (Li et al., 2020), foldback inversions were not recovered from the *exo1* mutant using the spontaneous GCR assay. The *exo1* single mutant was not tested in the DSB-induced GCR assay; however, *exo1* did reduce foldback inversions in the *sae2* background when a DSB was induced ~ 5kb from the target IR, consistent with the need for resection to expose ssDNA (Li et al., 2020). We observed a 4-fold higher frequency of survivors in the *exo1* mutant compared to WT, but the frequency was much lower than the *sae2* mutant (0.17% vs 9%) (Fig R1). Of the survivors, 16% had duplications as

Figure R1. Exo1 has a minor role in suppressing inverted duplications. Survival frequencies (A) and fraction of inverted duplications (B) of the indicated strains.

determined by qPCR. We observed a small but significant decrease in survival in the *sae2 exo1* double mutant relative to *sae2*, suggesting that Exo1-catalyzed end resection is partially responsible for exposing the IR. All tested survivors from the *sae2 exo1* double mutant had duplications, as measured by qPCR. Because we intend to follow up on the *exo1* results in future studies, we would prefer to omit these new data from the current submission.

Minor point:

In Figure 2A, the expected BamHI fragments at 1.2 and 11.2 kb are missing. Is there an explanation?

The 1.2 kb band (we changed the label to 1.3 kb to better represent 2 x 0.66 kb fragment) is present in all of the inverted duplicated clones and absent in the NHEJ clone, as expected. The 11.2 kb is only expected in the NHEJ clone, it's there, albeit faintly. Note that we modified Figure 2 to include all 12 clones analyzed by genomic DNA digest and Southern blot.

Reviewer #2

Previous studies have found that MRE11 and SAE2 suppress the formation of inverted duplications in budding yeast most likely by the cleavage of DNA hairpins formed from single-stranded inverted DNA sequences that are exposed following the resection of an adjacent DSB. This new paper from the Symington lab provides additional support for this model and highlights a role for the proofreading activity of Pol δ in stabilising the putative ssDNA hairpin by processing DNA flaps, as well as a potential role for Pol32 in converting the hairpin into a full-fledged hairpin capped chromosome. Prior studies have proposed that hairpin capped chromosomes are replicated to form dicentric isochromosomes that are either stabilized by the loss of one centromere or subsequently break during cell division leading to chromosome stabilization by a secondary DNA rearrangement. Consistent with these models, Al-Zain and colleagues identify both types of chromosomes amongst clones with inverted duplications induced by CRISPR/Cas9 DSBs.

Overall, this a nicely presented study that provides additional support for existing concepts and models that have been promoted and explored in other recent studies (e.g. Deng et al 2015; Li et al 2020).

Comments and questions:

1. It has been shown that Cas9 releases very slowly from DNA post-cleavage and therefore can inhibit subsequent DNA repair. It has also been shown that this stable post-cleavage complex can release the 3' end of the cleaved non-target strand, which can then be degraded by nucleases. Based on the information in Figure 1A, it would appear that the 3' end of the cleaved non-target strand is on the centromere proximal side of the DSB. Degradation of this end (possibly by Mre11) could lead to a foldback intermediate with a 5' heterologous DNA flap. This is just one example of how the specific properties of CRISPR/Cas9 DSBs could influence the subsequent DNA repair pathways and impinge on data interpretation. It would be prudent to determine to what extent, if any, the reported repair outcomes are influenced by the properties of the CRISPR/Cas9 DSB. As the authors point out in their Discussion (page 12), there are differences in the genetic requirements and frequencies of inverted duplications when comparing spontaneous with their DSB-induced events. How much of this difference is due to the nature of the CRISPR/Cas9 DSB? The authors should consider testing whether the gRNA target and non-target strands have any impact on the genetic dependencies of inversion deletion formation (i.e. whether the released 3' end is on the telomere or centromere side of the DSB), and whether an I-SceI DSB generates similar outcomes.

Although Cas9 releases very slowly from DNA post-cleavage *in vitro*, there is evidence for removal of Cas9 by the FACT complex *in vivo* (Wang et al., 2020), and in unpublished work from our lab we found efficient induction of ectopic recombination by Cas9. However, to address the reviewer's concern we replaced the binding site for gRNA-17 with a 36-bp HO cut site, which was designed to create a DSB 20

bp from the target IR, the same distance as generated by Cas9. We chose to use HO instead of I-SceI because it generally cuts with higher efficiency than I-SceI, and the short half-life of the protein means there are less issues with leaky expression. Since the DSB produced by HO can be repaired by NHEJ, we obtained the expected higher frequency of cell survival in the WT strain (0.33%), consistent with a previous study (Moore and Haber, 1996). Of these survivors, most had repaired by NHEJ (~80%). Remarkably, survival of *sae2* cells was almost 100-fold higher, reaching 26%. Similar to our observations with Cas9, the majority of *sae2* cells surviving HO induction formed small colonies indicative of inverted duplications and qPCR analysis confirmed this prediction. Thus, the large increase in DSB-induced inverted duplications observed in the *sae2* mutant reflects aberrant processing of a DSB and is not a consequence of Cas9 retention. These data are now presented in Figure S3 and Table S1.

2. I'm intrigued by the multiple Chromosome V bands detected in Figure 1D, which are particularly prevalent in the *sae2Δ* mutant. The authors attribute these bands to a heterogeneous population of cells that have undergone different rearrangements. Does this mean that the efficiency of DSB formation varies from cell to cell (i.e. not all cells experience a DSB before they divide) and/or that DSBs can be repaired faithfully so that the DNA can be subject to further rounds of cleavage? If so, then how do we know that the efficiency of DSB formation and/or faithful repair doesn't vary from strain to strain and affect survival frequency and the chances of detecting a particular GCR outcome? I also find it surprising that wild-type clone 2 and *sae2Δ* clone 7 (Figure 1D) have what appears to be an identical pattern of chromosome V bands. What are the chances that the same heterogeneous population of cells would occur in two different clonal populations from two different strains in such a small sample size?

Multiple bands can indicate two possibilities: a) a single parental cell with a dicentric chromosome that gives rise to a heterogeneous population of cells that break and repair the dicentric differently, and b) multiple cells sustaining independent initiating lesions which are then repaired differently. We think the first possibility is more likely for a number of reasons:

1. The multiple bands is a feature of cells with a dicentric chromosome and has been observed repeatedly in previous studies of spontaneous inverted duplications (Narayana et al., 2006; Deng et al., 2015), consistent with our findings.

2. The likelihood of two or more daughter cells from single lineage surviving a DSB is quite low. We suspect that the DSB often occurs after the cells have divided, and this could potentially affect survival frequency. The efficiency of DSB formation could vary between strains, but our past studies indicate that the efficiency of DSB formation is similar in WT and *sae2* strains. The likelihood of independent survivors from a clonal population (i.e. survivors from daughter cells that have independently sustained a DSB) is quite low by virtue of the fact that the survival frequency is already low to begin with. In *sae2* cells, if the measured survival frequency is around 10% of the plated number of cells, then the survival of any single daughter cells that sustain a DSB would be lower than 10%; it then follows that the likelihood that more than two daughters of a plated cell to survive is even lower, and certainly lower than the frequency with which we observe clones with multiple bands.

We agree that the band heterogeneity is similar between WT clone 2 and *sae2* clone 7. However, the distance migrated for some of the bands is slightly different between the two clones and there is an additional band in the *sae2* clone.

3. The authors mention that inversion between ER domains can eliminate Cas9 expression. These inversions account for ~20% of colonies growing on inducing conditions for wild-type cells. Were other strains checked for the occurrence of this rearrangement to exclude the possibility that this was contributing to survival frequency?

Cells that lose Cas9 expression grow faster when plated on β -estradiol-containing medium. We excluded fast growing colonies when counting colonies, as noted in the revised Methods section. When analyzing rearrangements, we rarely found colonies that had no rearrangement (and thus no DSB due to loss of Cas9 expression). The exception is the *pol32* mutant, as described in the manuscript.

4. Most of the Figures and figure legends do not include the sample size (n).

Survival and copy number data in the figures are dot plots, obviating the need to add the sample size. We added the sample size where this information was not apparent from the figure alone (Figures 2C and 3C).

Reviewer #3

In this study the authors investigate the effect of induced double strand breaks (DSBs) on the formation of Gross Chromosomal Rearrangements (GCRs). They observed that a DSB induced near a short inverted repeat results in the formation of inversion GCRs mediated by foldback inversions mediated by the short inverted repeat and that these GCRs are suppressed by Sae2 and the Mre11 complex. They provide extensive genetic analysis of this type of GCR formation along with detailed structural characterization using PCR, PFGE, qPCR and whole genome sequencing. Some of these methods are novel. And the analysis, such as mutagenesis of the initiating inverted repeat, is very thorough. In aggregate, the results provide a great deal of information about the mechanisms underlying the formation of inversion GCRs; these results, obtained in yeast, are of considerable relevance to the formation of these types of GCRs in cancer.

Overall, this is a very well-done study that is very thorough and uses a diversity of complementary, sometimes new, methods to generate data that support the conclusions presented. The paper is interesting and well written. Some questions about the experiments, data presentation and interpretations and conclusions are noted below, but overall, the conclusions seem solid. The results presented extend what we know about the formation of a particularly interesting class of GCRs, the inverted repeat mediated inversion GCRs. The most novel aspect of the study is the genetic analysis of DSB induced inversion GCRs which is in many ways the main focus of the paper; other than the analysis of Sae2, this analysis goes well beyond that of previous studies and is interesting and insightful.

Specific comments

Abstract. I think the abstract over-states some key results such as the supposed strong requirement in inversion formation such as the requirement of Pol delta Exo activity, POL32, RAD1 and RAD51 in different steps in GCR formation, which as note below are partial and sometimes more subtle than an absolute requirement. It also implies that less was previously known about inversion formation, even in the case of DSB induced events, than is the case. Regardless, the paper does describe new results. The abstract needs to be rewritten to provide a better description of the results present and their context.

We found that *pol3-01*, *pol32* and *rad51* mutations each reduce the survival frequency of the *sae2* mutant by >10-fold. Furthermore, the *pol3-01* mutation significantly changes the spectrum of events recovered from the *sae2* mutant. Thus, we feel that our conclusion on the role of Pol delta, as stated in the abstract, is accurate. We do not find a role for Rad1 in generation of inverted duplications; however, with the additional sequenced clones from the triple mutant it appears that the spectrum of secondary

events is altered.

P2-4. The introduction section is excellent. Informative and scholarly.

P4, L1. Ref 7 also carried out studies on this point so this is not correct. But obviously, this issue is a broad one worthy of multiple studies using different assays and target chromosomes, like the studies presented in the present paper.

We agree that DSB-induced inversions were analyzed in the Li et al. paper, as cited in our work. However, there were no data regarding the frequency of inverted duplications that initiate from a DSB in the Li et al study and the effect of the DSB on inverted duplications was only tested in *exo1* and *mus81* derivatives of *sae2*.

Fig 1 and others. A concern I have is how accurate a measurement is survival for the rate or efficiency of GCR formation. Related to this is what are adequate 100% and 0% GCR formation controls and how should relative survival data be presented. Absolute survival, as presented is useful. But also should survival relative to WT survival or relative to *sae2* mutant survival be presented? This is particularly true given that the different mutations studied could affect survival in ways unrelated to the rate of GCR formation. I don't doubt the conclusions, but possibly the limitations should be explained. I find the changes in the spectrum of GCRs formed under different experimental conditions to be useful and supportive of the authors conclusions; in this regard, the authors miss some opportunities to explain how GCR spectrum data support the conclusions from the survival data.

The frequency of cell survival in response to a targeted DSB is a standard read out for repair efficiency in the field (e.g., multiple papers from the Haber, Kupiec and Ira labs). It is also important to note that in our assay we monitored ability of cells to repair the DSB and not necessarily to form GCRs. This is particularly relevant to the *sae2* mutant, which shows an increased frequency of NHEJ at targeted DSBs relative to WT cells, and therefore we would not expect the survival of *sae2* cells deficient for activities that promote inverted duplication to show the same survival frequency in response to a targeted DSB as WT cells. Our finding that the greatly elevated survival frequency of *sae2* cells in response to an HO-induced DSB, and recovery of mostly inverted duplications, mirrors our results with Cas9 increases our confidence that survival frequency is an accurate measure of repair efficiency. We agree with the reviewer that the spectrum of events is often more informative than the frequency. This point is well illustrated by the finding that the *sae2* mutant exhibits only a ~10-fold increase in the rate of spontaneous GCRs, but nearly all the clones have inverted duplications (Putnam et al., 2014; Deng et al., 2015; Li et al., 2020). We found that *sae2* derivatives with a reduced fraction of survivors with inverted duplications showed a corresponding increase in NHEJ events (e.g., *sae2 pol3-01* and *sae2* with mutated target IR). As suggested by the reviewer, we have added a table to the revised manuscript that documents survival and inverted duplication frequencies relative to *sae2* for all the strains tested. In the revised manuscript, we emphasize how the spectrum of events is changed in the *sae2 pol3-01* double mutant relative to the *sae2* single mutant, reinforcing our conclusions from the survival data.

P6, L6-15 and Fig S2. This result would be easier to understand if Fig S2 better indicated with better precision where the P3/4 and P7/8 primers map. For example, could the primer sites be indicated on the CAN1 sequence in Fig S2D or a modified version. This would help explain the presence of P7/8 products in isolates that did not yield P3/4 products.

The positions of P3/P4 and P7/P8 have been added to Fig S2 (note: “Ns” fall towards the end of Sanger sequencing for the clones containing them).

P6, L21-23. There are some references that could be cited here regarding the formation of these types of rearrangements. The same is true for expanded chromosome sized rearrangements seen by PFG.

We apologize for this oversight and have added additional references.

P7, L2. Had likely repaired the DSB by NHEJ. Fig 2A does not provide enough information on the location of the Bam, EcoRV and Pst sites on the telomeric side of the DSB to allow drawing any mapping conclusions. I'm sure the authors could add this information.

The positions of the nearest restriction sites on the telomeric side of the break have been added (note, the position of the BamHI site on the centromeric side was corrected to 0.7 kb from 0.6 kb—the exact position is 664bp, and thus is rounded up).

P7, L26. Is this what the authors define as a long spacer in Table 1? The legend to Table 1 could be improved.

We agree with the Reviewer and have modified Table 1. Since, we recovered many more clones from the *pol3-01* background that used the inverted repeat including the gRNA PAM site, we have changed the name from long spacer to PAM IR.

P8, L21-32. The results show that the *pol3-01* mutation reduces survival of the *sae2* mutant but not to the level seen in wild-type and that a significant fraction of the survivors in the *sae2 pol3-01* mutant appear to be inversions. Shouldn't it be more clearly stated that this is consistent with a partial defect in flap processing? Given the partial defect caused by *pol3-01*, did the authors examine other nucleases?

Survival level in a strain with a *sae2* deletion is not expected to be reduced to WT levels if inverted duplications are eliminated since NHEJ is 10-fold higher in the absence of Sae2 (Lee and Lee, 2007; Deng et al., 2014). The *pol3-01* mutation reduces survival of *sae2* cells by 10-fold and of the survivors less than half have inverted duplications. Thus, the recovery of inverted duplications is reduced by ~20-fold, indicating that 95% of inverted duplications require the proofreading exonuclease activity of Pol delta. Furthermore, we found that half of the inverted duplications recovered from the *sae2 pol3-01* double mutant used the PAM IR a single nt from the DSB instead of the target IR. We tested the *sae2 mus81* double mutant but did not find a decrease in the survival frequency. Note, a recent study (Shultz and Jinks-Robertson, 2023) reported no effect of *rad1* or *mus81* mutations on the frequency of MMEJ repair of a targeted DSB, consistent with our finding that Rad1 has a minor role in flap processing involving short sequence homologies (Rad52-independent strand annealing).

P9, L1-6. Did the authors sequence any of the inversions seen in the *sae2 pol3-01* mutant as it appears that these can be recovered? This would test whether the *pol3-01* mutation alters the types of inversions seen. This is certainly what the comparison of *sae2 rad1* with *sae2 rad1 pol3-01* shows. This could be more clearly stated.

We thank the reviewer for suggesting that we sequence clones from the *sae2 pol3-01* double mutant. Of 19 clones sequenced, 9 used the target IR, 9 used the PAM IR, which would require removal of only a single nt prior to DNA synthesis, and the remaining clone used a different IR. We also sequenced 12

more clones from the *sae2 pol3-01 rad1* triple mutant and found that more than half of them used the PAM IR, not significantly different from the *sae2 pol3-01* double mutant, further evidence that Rad1 is not a back-up for Pol3 proofreading activity. Furthermore, we recovered additional inversion chromosome clones with a deleted centromere from the triple mutant. This observation is consistent with the need for Rad1 activity to promote secondary rearrangements between dispersed repeats.

P9, L23 to P10, L7. How does the effect of changing the DSB site compare to the results described in Ref 7 where different DSB sites were examined. Further, in regard to the conclusion about RAD1, didn't Ref 7 conclude that defects in Rad1-Rad10 caused little of no change in inversion frequency but did cause changes in the inversion sites used and thus played some role in flap processing and inversion formation.

In the revised manuscript, we cite the Li et al. (2020) paper where they also reported use of different IRs to the target IR when the DSB is generated further from the target IR. However, there are no data in the Li et al study on how the distance of the DSB from the target IR impacts frequency of events. Furthermore, Li et al. (2020) did not test the *sae2 rad1* double mutant in the DSB-induced assay, only in the spontaneous GCR assay where the location of the initiating lesion is unknown.

P10, L8-18 and Table 1. Having provided convincing evidence for the formation of several dicentric chromosomes. I'm not sure I'd title this section "evidence for a dicentric chromosome intermediate" unless there is convincing evidence that yet other rearrangements involved dicentric intermediates. Also, having called them dicentric, why then call them isochromosomes, especially in Table 1. Would "inversion chromosomes" or "dicentric inversion chromosomes" be better?

The inverted duplications associated with secondary rearrangements could in principle be formed without a dicentric chromosome intermediate. The recovery of 8 chromosome V inversion chromosomes (6 more were identified among the 12 additional clones sequenced from the *sae2Δ rad1Δ pol3-01* mutant) with a deletion removing a centromere is the strongest evidence we have for the formation of a dicentric chromosome intermediate. We moved this section to the end of the Results, after the description of secondary recombination events to stabilize the dicentric chromosome intermediate.

P10, L18 to P11, L11. I find this confusing. The authors are trying to argue that inversions in a *sae2* mutant require POL32 and RAD51 based on reduced survival in response to a DSB. There is reduced survival but not to WT survival levels. This is consistent with a partial requirement for POL32 and RAD51. Indeed, consistent with this, Fig 6E shows that inversions are formed in each case, obviously at different rates. I think the authors are pushing a view that POL32 is absolutely required when in fact the *pol32* defect is partial and more subtle in that the length of the inverted region is shorter, an interesting result that should be highlighted rather than obscured. This result is not as different from that of Ref 7 as the authors imply.

As noted above, the *sae2* mutant exhibits an increased frequency of NHEJ; thus, we would not expect a mutation that decreases inverted duplications in the *sae2* background to restore survival frequency to the WT level. We observed a 30-fold reduction in cell survival for the *sae2 rad51* mutant relative to *sae2* and a corresponding decrease in the frequency of inverted duplications (Table S1). For the *sae2 pol32* mutant, survival was reduced by 20-fold and there was an even greater decrease in the frequency of inverted duplications. Li et al., 2020 show a higher spontaneous foldback inversion rate in *pol32 sae2* cells and use that finding to argue that Pol32 (and BIR) does not initiate or resolve foldback inversions. It

is important to note that DSB-induced foldback inversions in *pol32 sae2* and *rad51 sae2* cells were not analyzed by Li et al. One issue that we discuss is if the frequency of spontaneous initiating lesions is higher in some genetic backgrounds, for example, *rad51*, then a decrease in the rate of spontaneous GCRs in the *sae2* background might not be apparent (more initiating lesions but less efficient repair). Along these lines, if there are more broken chromosome fragments in the *rad51* mutant then recombinant chromosomes could be stitched together by SSA, a *RAD51*-independent process.

P10, L26. I don't agree with this statement. What is reduced is duplication of the qPCR assay site, which is quite a bit centromeric from the inversion initiation site. Fig 6E and Table 1 actually show that the virtually all of the products recovered are inversions, albeit ones that duplicate a shorter region and don't duplicate the qPCR assay site. Again, an interesting result that is obscured by the desire to claim that POL32 is absolutely required.

All the inversions in Fig 6E duplicate the qPCR assay site. We don't see any evidence that the duplicated region is shorter in any of the clones analyzed, including *sae2 pol32*.

The qPCR assay is an interesting innovation in these studies. Related to the above point, the authors should point out the limitations of the assay related to the choice of locations for the qPCR primer sites as this is relevant to the interpretation of the data.

All inverted-repeat mediated foldback inversion on the left arm of Chr V described to date, as far as we can tell, including in Li et al., 2020, involved secondary rearrangements that used one of the following repeat sequences: a long terminal repeat (such as a delta element), a Ty element, a tRNA or a *PAU2*. There are no such repeat sequences telomeric to the qPCR primer site we chose, so we do not have a reason to believe that a substantial number of duplications would end before this site. The exceptions are rare events that resolve the dicentric by telomere addition, which would not depend on a repeat sequence. It is possible that we missed some clones with deletion of a centromere if these events removed one set of primer binding sites.

P12, L7. A dicentric intermediate is not necessarily involved. For example, strand switching during the inversion primed DNA synthesis could also account for the secondary rearrangements. All of the possibilities should be mentioned. Also, Ref 24 is relevant here.

We do cite other papers suggesting strand switching instead of by formation of a dicentric intermediate in the Discussion section. Since our data are more consistent with formation of a dicentric, we only consider this possibility in the model presented.

P12, L17. Ref 7 also extensively studied DSB induced inversions.

Li et al. (2020) did not examine the role of most of the mutations assessed here in their DSB-induced inversion assay.

P12, L21. Another explanation is that different substrates were used in the different studies, and that this effects genetic outcomes.

Addressed in the revised manuscript

P12, L21-24. Wasn't this proposed in Ref 7.

Ref 7 is cited

P14, L6-15. It should be noted that Ref 7 also extensively presented models involving dicentric chromosomes.

Ref 7 is cited for models involving dicentric chromosomes.

Could the Discussion be compacted a bit to focus on the key points.

We have reworked the Discussion as suggested.

REVIEWERS' COMMENTS

Reviewer #1 (Remarks to the Author):

This revised version addresses my comments.

Reviewer #2 (Remarks to the Author):

The authors' response and revisions have satisfactorily addressed my comments on the earlier version of their manuscript.

Reviewer #3 (Remarks to the Author):

I have reviewed the revised manuscript and the Author's responses to my initial comments. They have addressed my comments either through changes to the manuscript or by further explanation of their perspective. In this regard the manuscript has been significantly improved. I also think that the authors have similarly addressed the comments provided by the other reviewers, which were not as extensive as my comments. I have no further comments that I think would be productive.

Response to Reviewers

We are pleased that all three reviewers are satisfied with the revised manuscript and have no additional comments to be addressed.